# Metabolic profiling during malaria reveals the role of the aryl hydrocarbon receptor in regulating kidney injury

Michelle M Lissner[1], Katherine Cumnock[1], Nicole M Davis[1], José G Vilches-Moure[2], Priyanka Basak[1], Daniel J Navarrete[1], Jessica A Allen[3], David Schneider[1]*

[1]Department of Microbiology and Immunology, Stanford University, Stanford, United States; [2]Department of Comparative Medicine, Stanford University, Stanford, United States; [3]Division of Health, Mathematics and Science, Columbia College, Columbia, United States

**Abstract** Systemic metabolic reprogramming induced by infection exerts profound, pathogen-specific effects on infection outcome. Here, we detail the host immune and metabolic response during sickness and recovery in a mouse model of malaria. We describe extensive alterations in metabolism during acute infection, and identify increases in host-derived metabolites that signal through the aryl hydrocarbon receptor (AHR), a transcription factor with immunomodulatory functions. We find that $Ahr^{-/-}$ mice are more susceptible to malaria and develop high plasma heme and acute kidney injury. This phenotype is dependent on AHR in *Tek*-expressing radioresistant cells. Our findings identify a role for AHR in limiting tissue damage during malaria. Furthermore, this work demonstrates the critical role of host metabolism in surviving infection.

*For correspondence:
dschneid@stanford.edu

**Competing interests:** The authors declare that no competing interests exist.

## Introduction

Infection imposes metabolic challenges on hosts, including the generation of costly immune responses, repair of damaged tissues, and competition with pathogens for nutrients. Hosts cope with these pressures through systemic metabolic reprogramming, with varying effects on infection outcome. Metabolic alterations can be beneficial to hosts. For example, controlling circulating triglycerides and glucose minimizes tissue damage during sepsis (*Luan et al., 2019*; *Weis et al., 2017*); similarly, the switch from metabolic to immune transcriptional programs driven by the *Drosophila* transcription factor MEF2 during *Mycobacterium marinum* infection promotes survival (*Clark et al., 2013*). In contrast, some metabolic changes are detrimental to hosts, such as reduced insulin signaling in *M. marinum*-infected fruit flies, leading to wasting and death (*Dionne et al., 2006*). Metabolic changes have infection-specific outcomes; for example, infection-induced anorexia promotes survival in some infection contexts, but not others (*Ayres and Schneider, 2009*; *Cumnock et al., 2018*; *Rao et al., 2017*; *Wang et al., 2016*; *Wang et al., 2018*). Metabolic changes can mediate these pathogen-specific outcomes through two distinct host protection strategies: by affecting either pathogen killing, a process called resistance, and/or the degree of collateral damage to the host per microbe, called disease tolerance (*Ayres and Schneider, 2009*). Better understanding of how systemic infection-induced metabolic changes affect infection will inform therapeutics that intentionally alter metabolism to improve outcomes.

Over 400,000 people die annually from malaria, which is caused by mosquito-transmitted *Plasmodium* parasites (*WHO, 2018*). The effect of host metabolism on malaria outcome is poorly understood. Metabolic changes such as lactic acidosis occur during malaria (*Miller et al., 2013*) and a number of studies have reported metabolomics on *in vivo Plasmodium* infection (*Abdelrazig et al.,*

*2017*; *Gardinassi et al., 2017*; *Gardinassi et al., 2018*; *Ghosh et al., 2012*; *Ghosh et al., 2016*; *Gupta et al., 2017*; *Lee et al., 2014*). Nevertheless, these experiments have important limitations. Field studies often do not capture events in early infection that occur prior to the onset of clinical symptoms. Furthermore, important metabolic alterations may occur on the scale of days, which would require the collection of densely spaced samples often infeasible outside of lab experiments.

Malaria can lead to pathology including severe anemia and acute kidney injury (AKI) (*Haldar and Mohandas, 2009*; *Koopmans et al., 2015*; *Trang et al., 1992*). During blood stage infection, parasites infect, proliferate within, and lyse red blood cells (RBCs). This hemolysis releases hemoglobin and then heme into plasma (*Miller et al., 2013*). Free heme can catalyze the formation of reactive oxygen species, damaging cells and tissues including the kidneys (*Chiabrando et al., 2014*; *Tracz et al., 2007*). To mitigate heme toxicity, heme levels are regulated in plasma by heme-binding proteins and intracellularly by the heme degradation enzyme heme oxygenase-1 (HO-1), among others (*Chiabrando et al., 2014*). During malaria, these mechanisms limit AKI (*Ramos et al., 2019*; *Seixas et al., 2009*). Clinically, the development of AKI during malaria correlates with high heme levels (*Elphinstone et al., 2016*; *Plewes et al., 2017*). Together, these data suggest that heme toxicity is an important cause of tissue damage during malaria.

Heme metabolism produces agonists of the aryl hydrocarbon receptor (AHR), a nuclear receptor transcription factor. AHR ligands include the environmental toxin 2,3,7,8-tetrachlorodibenzo-*p*-dioxin (TCDD), the heme metabolites bilirubin and biliverdin, tryptophan metabolites including kynurenine, and indoles produced by the commensal microbiota (*Rothhammer and Quintana, 2019*; *Stockinger et al., 2014*). AHR functions in a ligand- and cell-specific manner in processes such as the regulation of T cell differentiation, homeostasis in barrier tissues, and cell proliferation (*Esser and Rannug, 2015*; *Rothhammer and Quintana, 2019*; *Stockinger et al., 2014*). In the absence of AHR, mice are more susceptible to a variety of infections and inflammatory insults, including *Plasmodium berghei* ANKA, a rodent malaria parasite that causes lethal cerebral malaria (*Bessede et al., 2014*; *Brant et al., 2014*; *Di Meglio et al., 2014*; *Kimura et al., 2014*; *Moura-Alves et al., 2014*; *Sanchez et al., 2010*; *Shi et al., 2007*). As AHR has diverse functions in many cell types, including immune cells, endothelial cells, and hepatocytes, its precise role in a given infection is often undefined (*Agbor et al., 2011*; *Boule et al., 2018*; *Chen et al., 2019*; *Chng et al., 2016*; *Jux et al., 2011*; *Metidji et al., 2018*; *Sekine et al., 2009*; *Walisser et al., 2005*).

Here, we use metabolomics to systemically characterize metabolism in C57BL/6 mice infected with *P. chabaudi* (*Pc*). *Pc* is a rodent malarial parasite that recapitulates several key features of human malaria, including systemic inflammation and nonlethal anemia during acute infection, but does not cause severe complications like AKI in wild-type C57BL/6 mice (*Stephens et al., 2012*). We find that AHR ligands are most abundant during acute sickness and demonstrate that AHR signaling is an essential host protection mechanism to regulate plasma heme, limit kidney damage, and promote survival during malaria. Moreover, we determine that these functions are dependent on AHR signaling in *Tek*-expressing radioresistant cells.

## Results

### Malaria is characterized by stages with unique immune, metabolic, and tissue damage events

To identify connections between host metabolism and malaria pathogenesis, we first asked how well-known features of *Pc* malaria temporally relate to one another. We collected blood and plasma from *Pc*-infected C57BL/6 mice daily from 0 to 25 days post-infection (DPI) as well as mock-infected control mice on 5, 7, 10, 12, 15, 19, and 25 DPI. We divided this time series into early, acute, and late infection based on parasitemia; while early infection was marked by undetectable parasitemia, parasites infected up to one-third of RBCs during the acute stage, with a small recrudescence in parasitemia occurring during late infection (*Figure 1A*). As parasitemia rose during acute infection, mice developed pathology including liver damage, indicated by plasma levels of alanine aminotransferase (ALT), and anemia, which was not ameliorated until late infection (*Figure 1B*). We next evaluated the immune response throughout infection by analyzing peripheral blood for pro-inflammatory cells and cytokines with known functions in malaria pathogenesis and pathology (reviewed in *Aitken et al., 2018*; *Angulo and Fresno, 2002*; *Dunst et al., 2017*; *Wolf et al., 2017*). We observed that aspects

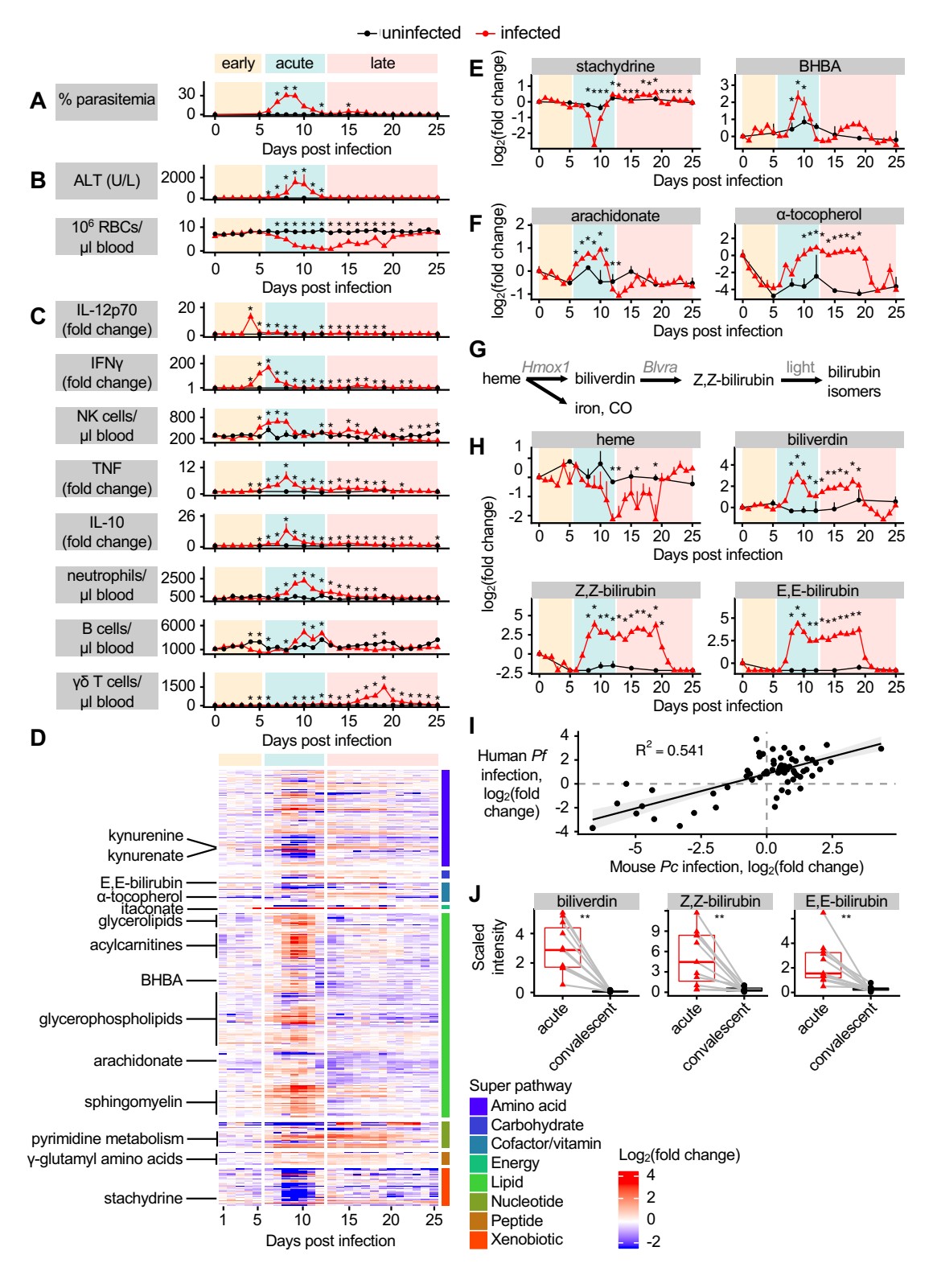

**Figure 1.** Dynamic multi-omic profiling of *Pc*-infected mice reveals broad immune and metabolic changes. (A) Parasitemia, (B) liver damage and anemia, and (C) selected peripheral blood immune cells and cytokines during 25 days of malaria. (D) 370 metabolites with altered scaled intensity in plasma during malaria, arranged by super pathway. Fold change of scaled intensity of (E) stachydrine, BHBA, (F) arachidonate, and α-tocopherol in plasma during malaria, relative to day 0. (G) Schematic of metabolites and genes of heme metabolism. (H) Fold change of scaled intensity of heme-

*Figure 1 continued on next page*

*Figure 1 continued*

related metabolites in plasma during malaria, relative to day 0. (I) Fold change of metabolites (n = 77) that were significantly altered in the plasma of both *Pc*-infected mice and pediatric cerebral malaria patients, plotted by fold change relative to day 0 samples for mice and relative to convalescent values for patients. Data are fitted with a linear model. (J) Scaled intensity of heme-related metabolites during human malaria (n = 11 patients per condition). In A-C, E, F, and H, data are presented as mean + SEM and p-values were determined by comparing each infected time point to all uninfected values using two-way ANOVA with FDR correction (n = 5 mice on 0 DPI, five infected mice each day, and two uninfected mice each day). *p<0.05. In J, p-values were determined using a Wilcox test. **p<0.01. These experiments were performed once.

The online version of this article includes the following source data and figure supplement(s) for figure 1:

**Source data 1.** Source data for *Figure 1*.
**Figure supplement 1.** Gating strategy used to define blood cell populations.
**Figure supplement 2.** Production of heme metabolites during malaria.
**Figure supplement 2—source data 1.** Source data for *Figure 1—figure supplement 2*.

of the immune response were activated prior to the onset of parasitemia and pathology. Interleukin 12 (IL-12p70) and interferon γ (IFNγ) increased in blood during early infection; acute infection was marked by elevated circulating natural killer (NK) cells, neutrophils, and B cells, as well as increased tumor necrosis factor (TNF) and IL-10 (*Figure 1C*, *Figure 1—figure supplement 1*). Circulating γδ T cells increased in peripheral blood during late malaria, when they control recrudescence (*Mamedov et al., 2018*; *Figure 1C*). This analysis revealed that each day of infection is marked by a unique combination of immune and pathological events, with the immune response both predating and outlasting the parasitemia and pathology of acute infection. Moreover, we established a time-line of many well-understood features of malaria to provide context to our metabolomic analysis.

We next asked how host metabolism changes during each stage of malaria. Untargeted metabolomics identified 587 metabolites in plasma. We selected metabolites whose maximum or minimum scaled intensity was (a) at least twofold changed from baseline and (b) significantly different from uninfected samples (p<0.05 by t-test with FDR correction). Of the 370 metabolites we identified as altered according to these stringent criteria, 66% increased in scaled intensity during *Pc* malaria, most during acute infection (*Figure 1D*). These metabolites illustrate the significant alterations to host energy metabolism that occur during infection (*Cumnock et al., 2018*). For example, broad classes of lipids increased during acute infection, including acylcarnitines, glycerolipids, glycerophospholipids, and sphingomyelin (*Figure 1D*). Metabolites associated with food, including stachydrine, a glycine betaine analog found in grains (*Filipčev et al., 2018*), decreased in scaled intensity during acute infection (*Figure 1E*), indicative of reduced food intake. Ketone bodies also increased in scaled intensity during acute infection (*Figure 1E*), suggesting that mice entered ketosis, perhaps linked to decreases in food consumption (*Cumnock et al., 2018*). Acute infection was also characterized by increased scaled intensity of inflammation-induced metabolites like arachidonate, followed by antiinflammatory metabolites like α-tocopherol, a form of vitamin E, in acute and late infection (*Figure 1F*). Several γ-glutamyl amino acids increased in scaled intensity during acute and late infection (*Figure 1D*), suggesting flux through the γ-glutamyl cycle and potentially glutathione biogenesis. A subset of metabolites related to pyrimidine metabolism also increased in scaled intensity during acute and late infection (*Figure 1D*). Overall, this analysis showed that metabolic changes occurred primarily during acute infection, with relatively few changing only during early or late infection.

We were curious about the pathological and metabolic implications of hemolysis during malaria, which decreases circulating RBCs to just 10% of baseline levels (*Figure 1B*). Hemolysis releases heme into plasma, where it is bound by heme scavengers, imported into cells in heme-metabolizing organs including liver and kidney, and metabolized into biliverdin by heme oxygenase 1 (HO-1), then into bilirubin by biliverdin reductase (BVR) (*Chiabrando et al., 2014*). Light exposure converts Z,Z-bilirubin into isomers including E,E-bilirubin (*Figure 1G*; *Rehak et al., 2008*). The scaled intensity of heme remained constant during most of acute infection (*Figure 1H*) despite significant hemolysis (*Figure 1B*), indicating activation of systemic heme metabolism. However, biliverdin, Z,Z-bilirubin, and E,E-bilirubin increased in scaled intensity during acute and late infection (*Figure 1H*). This is consistent with increased transcription of *Hmox1* and *Blvra*, the genes encoding HO-1 and BVR, in the liver during acute infection (*Figure 1—figure supplement 2A*), as well as HO-1 activation during

malaria (*Ramos et al., 2019*; *Seixas et al., 2009*). Therefore, acute *Pc* malaria is characterized by stable plasma heme levels combined with an increase in heme breakdown products.

To determine how the metabolic changes we observed in our mouse malaria model compare to human malaria, we analyzed a metabolomics dataset describing the plasma of *P. falciparum*-infected pediatric patients during acute cerebral malaria (CM) and 30 days post-treatment (*Gupta et al., 2017*). While uncomplicated *Pc* malaria in mice and CM caused by *P. falciparum* in humans differ substantially in lethality and pathogenesis, the two forms of malaria do share similar features, such as massive RBC destruction and systemic immune activation. We identified 77 metabolites with significantly altered intensity in both acute CM and *Pc* malaria, and observed similar magnitude and directionality of malaria-induced changes (p-value=1.572e-14) (*Figure 1I*). For example, increased scaled intensity of ketone bodies (BHBA) was observed in CM (*Figure 1—figure supplement 2B*), just as in *Pc* malaria (*Figure 1E*), suggesting that both humans and mice enter ketosis during acute infection. Like in *Pc* malaria, acute CM was characterized by increased scaled intensity of inflammation-induced arachidonate, and a trend toward decreased intensity of food-derived stachydrine (*Figure 1—figure supplement 2B*). As in *Pc* malaria, the scaled intensity of biliverdin and both measured bilirubin isomers was elevated during acute CM (*Figure 1J*), while heme scaled intensity was not different between the acute and convalescent timepoints in the clinical samples (*Figure 1—figure supplement 2B*).

These data identify distinct malaria stages with unique physiological, immune, and metabolic events. Comparison of the plasma metabolome of clinical CM and mouse *Pc* malaria suggests that *Pc* infection recapitulates many metabolic changes observed clinically, including increases in heme metabolites during acute infection.

## AHR ligands are more abundant during acute infection

Biliverdin and bilirubin activate AHR, an immunomodulatory transcription factor with agonists of dietary, microbial, and host origin (*Rothhammer and Quintana, 2019*; *Stockinger et al., 2014*). We therefore looked for other AHR ligands in our dataset. Tryptophan catabolism via the kynurenine pathway (*Figure 2A*) produces metabolites with varying affinities for AHR, including L-kynurenine, kynurenate, 3-hydroxy-DL-kynurenine, and quinolinic acid (*Desvignes and Ernst, 2009*; *Rothhammer and Quintana, 2019*). Tryptophan decreased in scaled intensity during acute infection, while downstream products including kynurenine, kynurenate, and quinolinate increased (*Figure 2B*), as observed previously in a *P. berghei* ANKA infection (*Clark et al., 2005*). *Pc* infection resulted in increased expression of *Ido1* and decreased expression of downstream kynurenine pathway genes in the liver (*Figure 2—figure supplement 1A*), suggesting that these metabolites may be produced in other tissues during malaria. Moreover, in cerebral malaria patients, the scaled intensity of tryptophan, kynurenine, and kynurenate was elevated during acute infection relative to convalescence (*Figure 2C*).

AHR ligands also include indoles, a class of metabolites derived from microbial metabolism of tryptophan (*Rothhammer and Quintana, 2019*). By untargeted metabolomics, the scaled intensity of indoleacetate trended towards decreasing during acute infection (*Figure 2—figure supplement 1B*) consistent with infection-induced anorexia during this time limiting energy input for both the host and microbiota (*Figure 1E*; *Cumnock et al., 2018*). Other AHR ligands, including the arachidonic acid derivatives PGE2, leukotriene B4, and lipoxin A4, were not measured in our metabolomics experiment (*Denison and Nagy, 2003*; *Rothhammer and Quintana, 2019*).

To confirm our metabolomics data, we quantified the plasma levels of select AHR ligands at 9 DPI in infected and uninfected mice. As in our metabolomics data, the concentration of plasma bilirubin was elevated during acute infection (*Figure 2D*). We measured tryptophan metabolites by targeted mass spectrometry and confirmed that plasma tryptophan decreased during infection, whereas the concentration of L-kynurenine, 3-hydroxy-DL-kynurenine, and quinolinic acid increased (*Figure 2E*). During acute malaria, bilirubin reached higher plasma concentration than any tryptophan metabolite by at least one order of magnitude. In summary, we observed that AHR ligands derived from heme and tryptophan metabolism increased in plasma of mice and humans during acute *Plasmodium* infection.

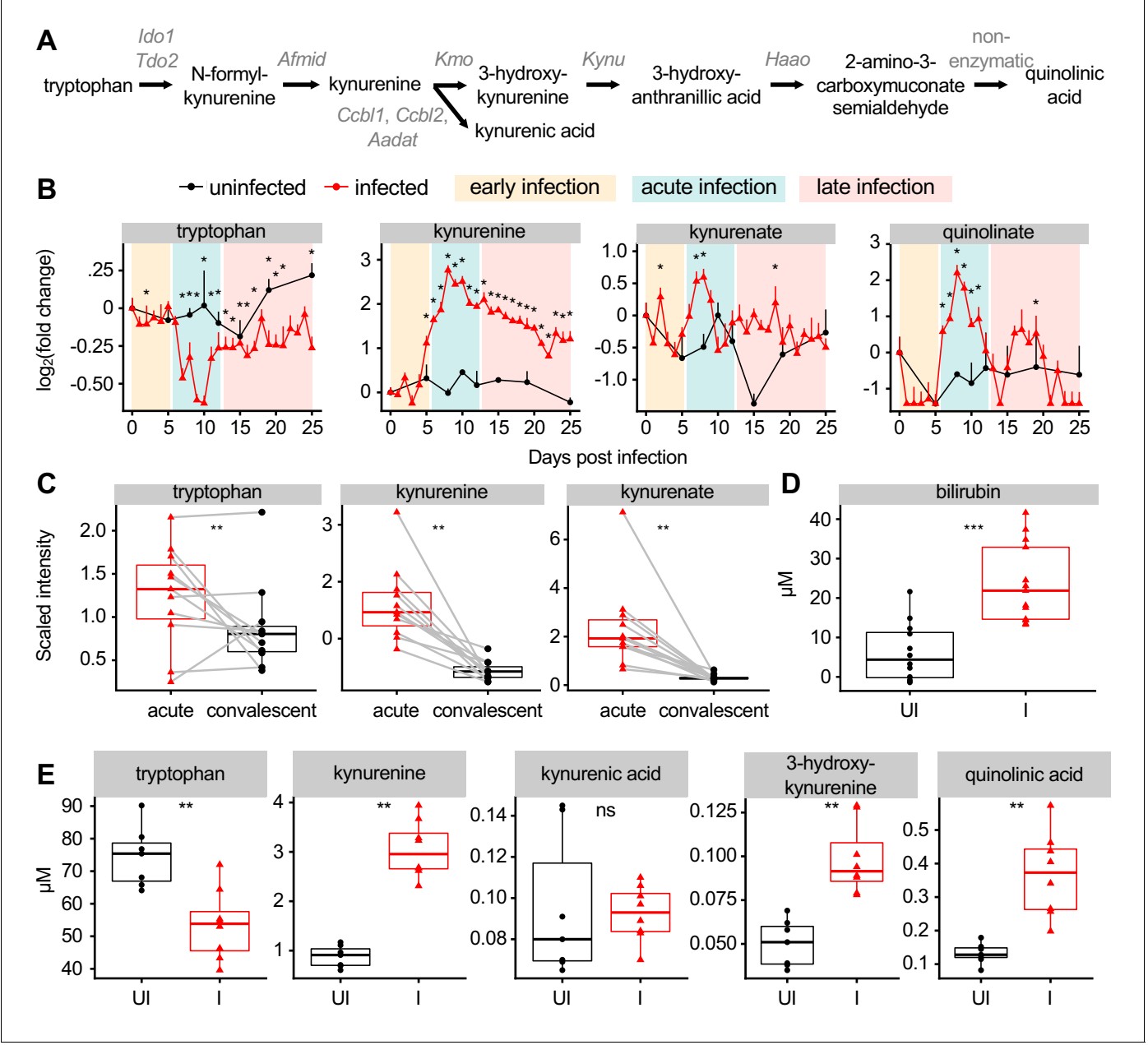

**Figure 2.** Malaria modulates AHR ligands in the plasma of mice and patients. (A) Schematic of metabolites and genes of the kynurenine pathway. (B) Fold change of scaled intensity of kynurenine pathway compounds during *Pc* infection, relative to day 0 (mean + SEM). p-Values were determined by comparing each infected time point to all uninfected values using two-way ANOVA with FDR correction (n = 5 mice on 0 DPI, five infected mice each day, and two uninfected mice each day). *p<0.05. (C) Scaled intensity of kynurenine pathway compounds in pediatric cerebral malaria patients (n = 11 patients per condition). (D) Quantification of bilirubin (n = 12–13 mice per condition) and (E) kynurenine pathway metabolites (n = 5–6 mice per condition) at 9 days post *Pc* infection. p-Values were determined in C-E using a Wilcox test. *p<0.05, **p<0.01, ***p<0.001, ****p<0.0001. These experiments were performed once.

The online version of this article includes the following source data and figure supplement(s) for figure 2:

**Source data 1.** Source data for *Figure 2*.
**Figure supplement 1.** Production of AHR ligands during malaria.
**Figure supplement 1—source data 1.** Source data for *Figure 2—figure supplement 1*.

## AHR signaling is protective during *Pc* infection

Given that AHR ligands were elevated in plasma during acute infection, we hypothesized that AHR signaling affects the outcome of *Pc* infection. We infected female littermate $Ahr^{+/+}$, $Ahr^{+/-}$, and $Ahr^{-/-}$ mice with *Pc* and monitored parasite load and survival over a 15-day time course that captured both acute infection and recovery. $Ahr^{-/-}$ mice developed higher parasitemia than $Ahr^{+/+}$ and $Ahr^{+/-}$ mice (*Figure 3A*). All genotypes had similar maximum parasite density, although parasite density in $Ahr^{-/-}$ mice peaked a day earlier (*Figure 3B*). $Ahr^{-/-}$ mice developed more severe anemia, indicated by RBC density (*Figure 3C*), at least partly due to increased parasite-mediated hemolysis. $Ahr^{-/-}$ mice also had more severe sickness as measured by weight loss and temperature decrease (*Figure 3D–E*). While most $Ahr^{+/+}$ and $Ahr^{+/-}$ mice survived, all $Ahr^{-/-}$ mice succumbed to infection between days 9 and 12 (*Figure 3F*). Because $Ahr^{+/+}$ and $Ahr^{+/-}$ mice had equivalent disease severity, we used either $Ahr^{+/+}$ or $Ahr^{+/-}$ mice as controls for subsequent experiments. We observed similar, if less pronounced, trends for male $Ahr^{+/+}$, $Ahr^{+/-}$, and $Ahr^{-/-}$ mice (*Figure 3—figure supplement 1*).

Our metabolomics data suggested that AHR signaling during malaria could be activated by tryptophan and/or heme metabolites. To ask whether susceptibility to malaria is dependent on tryptophan metabolism, we measured kynurenine and other tryptophan metabolites in *Pc*-infected mice lacking indoleamine 2,3-dioxygenase 1 (*Ido1*), one of two enzymes that catalyzes the rate-limiting step of the tryptophan pathway (*Figure 2A*; *Fatokun et al., 2013*). The concentrations of kynurenine and 3-hydroxy-kynurenine in $Ido1^{-/-}$ mice at 9 DPI were lower than infected $Ido1^{+/+}$ mice and statistically indistinguishable from uninfected mice of either genotype (*Figure 3—figure supplement 2A*). Nevertheless, $Ido1^{-/-}$ and $Ido1^{+/+}$ mice had similar parasite load and survival (*Figure 3—figure supplement 2B–D*). We conclude that AHR signaling, but not by kynurenine pathway metabolites, is required during *Pc* infection under our experimental conditions.

## $Ahr^{-/-}$ mice develop AKI during malaria

To understand why $Ahr^{-/-}$ mice were more susceptible to malaria, we asked which tissues were specifically injured in infected $Ahr^{-/-}$ mice compared to $Ahr^{+/-}$ mice. Liver damage is a common

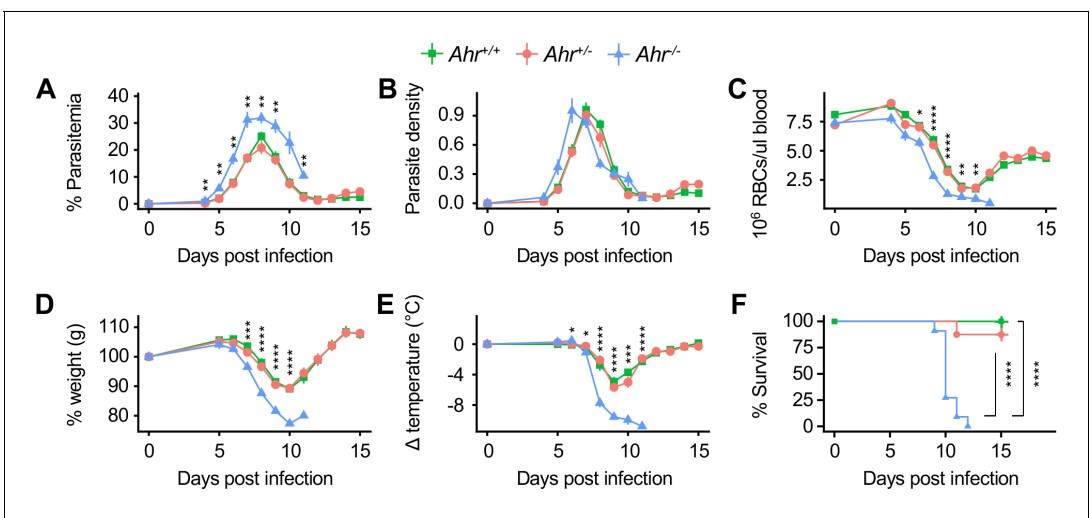

**Figure 3.** $Ahr^{-/-}$ mice are susceptible to malaria. (A) Parasitemia, (B) parasite density, (C) RBCs/µl blood, change in (D) body weight and (E) temperature relative to day 0, and (F) survival of *Pc*-infected $Ahr^{+/+}$, $Ahr^{+/-}$ and $Ahr^{-/-}$ mice (n = 10, 8, and 11, respectively). p-Values in A-E were determined using two-way ANOVA with FDR correction comparing $Ahr^{+/-}$ and $Ahr^{-/-}$ mice; values are mean ± SEM. p-Values in F were determined using a log-rank test. *p<0.05, **p<0.01, ***p<0.001, ****p<0.0001. Data were combined from three independent experiments.

The online version of this article includes the following source data and figure supplement(s) for figure 3:

**Source data 1.** Source data for *Figure 3*.
**Figure supplement 1.** Male*Ahr-/-*mice are susceptible to malaria.
**Figure supplement 1—source data 1.** Source data for *Figure 3—figure supplement 1*.
**Figure supplement 2.** *Ido1-/-*mice are not susceptible to malaria.
**Figure supplement 2—source data 1.** Source data for *Figure 3—figure supplement 2*.

pathology during malaria, and *Ahr*[+/-] mice had liver damage at 8 DPI, indicated by a 100-fold increase in plasma levels of alanine aminotransferase (ALT) relative to baseline (*Figure 4A*). Plasma ALT in *Ahr*[-/-] mice increased by 8-fold above baseline. Additionally, livers from *Pc*-infected *Ahr*[+/-] mice had more frequent histological evidence of parenchymal necrosis than *Ahr*[-/-] mice, although both genotypes showed similar extent of inflammation and margination of leukocytes along the vascular endothelium (*Figure 4—figure supplement 1A*). Thus, *Ahr*[-/-] mice develop less severe liver damage than *Ahr*[+/-] mice during *Pc* malaria. Histological analysis of spleens revealed comparable

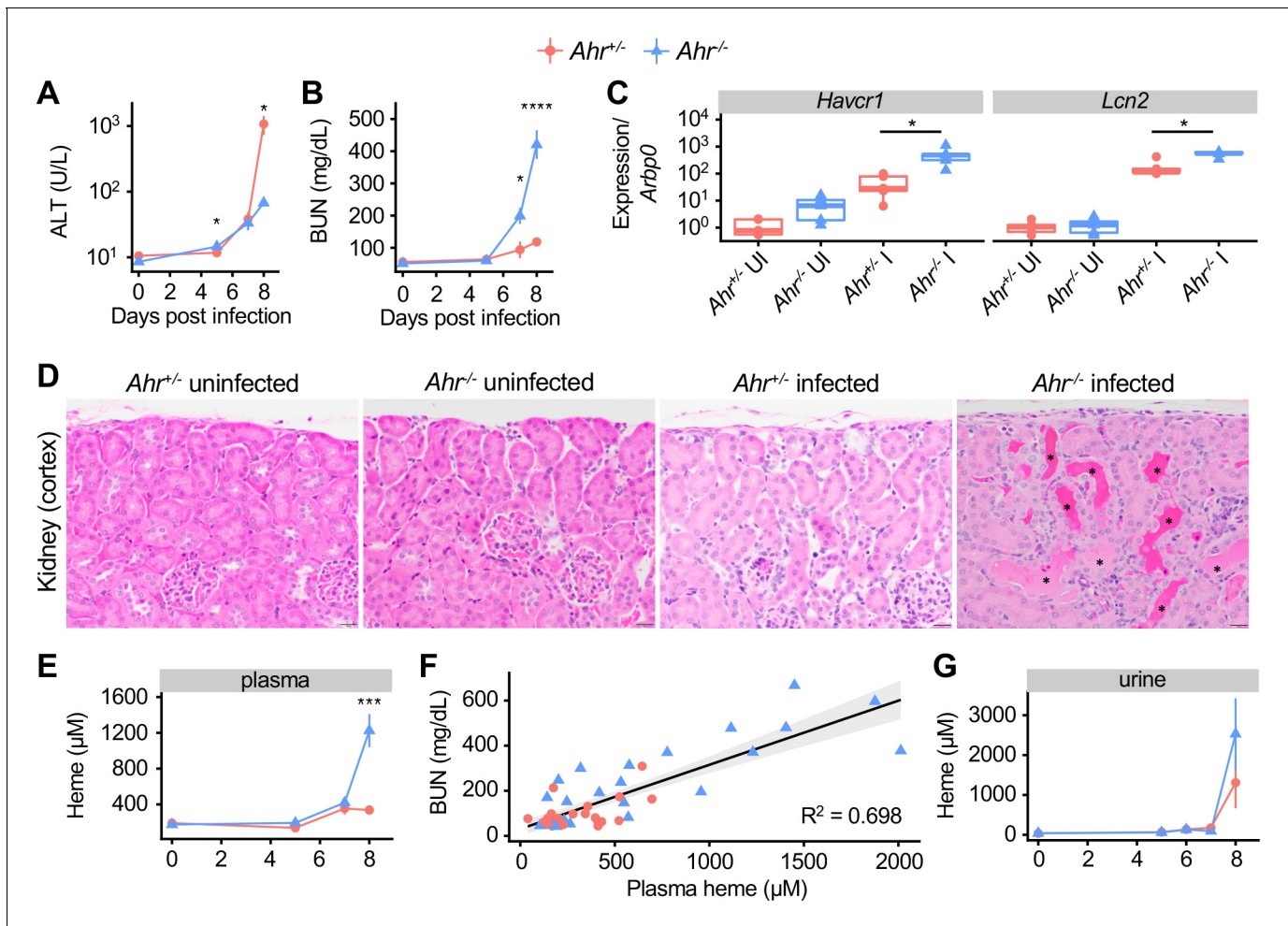

**Figure 4.** Acute kidney injury and inappropriate heme regulation in *Ahr*[-/-] mice during malaria. (A) ALT and (B) BUN in plasma of *Ahr*[+/-] and *Ahr*[-/-] mice during infection (n = 3–9 per group). (C) Gene expression in kidneys from *Ahr*[+/-] and *Ahr*[-/-] mice on 8 DPI (normalized to *Arbp0* using the ddCT method, n = 5 per condition). (D) Representative images of H and E-stained kidney tissue from *Ahr*[+/-] and *Ahr*[-/-] mice (magnification: 40x, scale bar: 20 μm). Black asterisks indicate dilated renal tubules with eosinophilic proteinaceous fluid. (E) Total heme in plasma of *Pc*-infected *Ahr*[+/-] and *Ahr*[-/-] mice (n = 3–9 per group). (F) Correlation of heme and BUN from individual mice (also plotted in B and E) and fit with a linear model. (G) Total heme in urine of *Pc*-infected *Ahr*[+/-] and *Ahr*[-/-] mice (n = 3–8 per group). p-Values in A, B, E, and G were determined using two-way ANOVA with FDR correction; values are mean ± SEM. p-Values in C were determined using a Wilcox test. *p<0.05, **p<0.01, ***p<0.001, ****p<0.0001. Each timepoint was collected in one to two independent experiments.

The online version of this article includes the following source data and figure supplement(s) for figure 4:

**Source data 1.** Source data for *Figure 4*.

**Figure supplement 1.** Representative images of H and E-stained tissue showing (A) liver, (B) spleen, and (C) lung, and (D) kidney arcuate vein from *Ahr*[+/-] and *Ahr*[-/-] mice (magnification: 40x).

**Figure supplement 2.** Heme metabolism appears largely normal inAhr-/-mice during malaria.

**Figure supplement 2—source data 1.** Source data for *Figure 4—figure supplement 2*.

**Figure supplement 3.** Control andAhr-/-mice are equally susceptible to phenylhydrazine-induced heme overload.

**Figure supplement 3—source data 1.** Source data for *Figure 4—figure supplement 3*.

red pulp extramedullary hematopoiesis and vascular leukocyte margination in both genotypes during infection (*Figure 4—figure supplement 1B*), while lungs from infected and uninfected *Ahr^+/-* and *Ahr^-/-* mice were within normal limits (*Figure 4—figure supplement 1C*).

To evaluate kidney function, we measured blood urea nitrogen (BUN) levels. At baseline, mice of both genotypes had comparable BUN. During acute infection, BUN in *Ahr^+/-* mice increased by two-fold above baseline, indicative of a mild to moderate decrease in kidney function; in *Ahr^-/-* mice, BUN increased by eightfold, indicative of a substantial decrease in function (*Figure 4B*; *Ramos et al., 2019*; *Wei and Dong, 2012*). The genes *Kidney injury molecule 1* (*Havcr1*) and *Lipocalin 2* (*Lcn2*), markers of kidney injury, were induced in both genotypes by *Pc* infection, and to significantly higher levels in *Ahr^-/-* mice at 8 DPI (*Figure 4C*; *Han et al., 2002*; *Moschen et al., 2017*). Histologically, kidneys from *Ahr^-/-* mice had evidence of leukocyte margination (most prominently in the arcuate vessels), frequent cortical tubular dilation (with or without luminal protein), and rare tubular epithelial cell necrosis with luminal sloughing, whereas kidneys from *Ahr^+/-* mice only show evidence of leukocyte margination (*Figure 4D*, *Figure 4—figure supplement 1D*). Together, these data show that *Ahr^+/-* mice have mild kidney injury during *Pc* malaria, whereas *Ahr^-/-* mice develop AKI.

We next aimed to determine the cause of kidney pathology in *Ahr^-/-* mice. Parasites sequestered in capillaries could obstruct vessels and cause kidney damage. To test this hypothesis, we measured expression of *Merozoite surface protein 1* (*Msp1*) in perfused kidney tissue as a proxy for parasite abundance. We observed no difference in *Msp1* gene expression in the kidneys of *Ahr^+/+* and *Ahr^-/-* mice (*Figure 4—figure supplement 2A*), suggesting that increased parasite sequestration within *Ahr^-/-* kidneys was not responsible for kidney injury. Kidneys can also sustain injury when exposed to free heme; during malaria and other hemolytic conditions, lysed RBCs release heme into plasma, and mice with impaired heme metabolism can suffer from heme-mediated AKI (*Ramos et al., 2019*; *Seixas et al., 2009*; *Vinchi et al., 2013*; *Zarjou et al., 2013*). Plasma heme levels in *Ahr^+/-* and *Ahr^-/-* mice were equivalent at baseline; however, at 8 DPI, plasma heme in *Ahr^-/-* mice was elevated sevenfold above baseline, compared to 1.8-fold in *Ahr^+/-* mice (*Figure 4E*). These findings are consistent with a previous observation of increased serum iron in *P. berghei* ANKA-infected *Ahr^-/-* mice (*Brant et al., 2014*). These data also demonstrate that kidney function during *Pc* infection is correlated with heme concentration. Plotting plasma heme and BUN values for 49 *Ahr^+/-* and *Ahr^-/-* mice on 0, 5, 7, and 8 dpi reveals a correlation with adjusted $R^2$ = 0.694 (p-value=$6.699 \times 10^{-14}$) (*Figure 4F*). We conclude that *Pc*-infected *Ahr^-/-* mice suffered from AKI, likely caused by heme toxicity.

Normal heme metabolism involves binding free heme in plasma with scavengers, intracellular import, and HO-1-mediated breakdown in the liver and other heme-metabolizing organs *Chiabrando et al., 2014*; during malaria, heme is also excreted in urine (*Ramos et al., 2019*). The elevated plasma heme observed in *Pc*-infected *Ahr^-/-* mice could be caused by excess heme release into plasma and/or issues with heme transport and breakdown subsequent to release in plasma. To discriminate between these possibilities, we measured urinary heme and found that *Ahr^+/-* and *Ahr^-/-* mice had similarly elevated heme levels in urine (*Figure 4G*). To test for inappropriate iron deposits in the kidneys of *Ahr^-/-* mice, we performed Perls Prussian blue staining, which revealed similar accumulation of iron-containing compounds in the cortical tubular epithelial cells and within the tubular lumina of both *Ahr^+/-* and *Ahr^-/-* infected mice (*Figure 4—figure supplement 2B*). We also evaluated levels of HO-1 and ferritin heavy chain (FTH), two key proteins involved in heme metabolism required for survival during malaria (*Ramos et al., 2019*), as well as the heme transporters divalent metal transporter 1 (DMT1) and heme carrier protein 1 (HCP1). We did not observe a defect in protein levels in *Ahr^-/-* mice (*Figure 4—figure supplement 2C*). Transcriptional analysis of an expanded set of heme metabolism genes in liver revealed comparable expression of most genes in infected *Ahr^+/-* and *Ahr^-/-* mice, with the notable exceptions of *Heme responsive gene 1* (*Hrg1* or *Slc48a1*) and *Solute carrier family 40 member 1* (*Slc40a1* or *Ferroportin*), which remained at baseline levels in infected *Ahr^-/-* mice (*Figure 4—figure supplement 2D*). Lastly, to determine if AHR is required to control plasma heme in other hemolytic conditions, we employed a model of phenylhydrazine-induced hemolysis (*Dutra et al., 2014*). *Ahr^+/-* and *Ahr^-/-* mice challenged with phenylhydrazine had equivalent survival, hemolysis, plasma heme, and BUN (*Figure 4—figure supplement 3A–D*). Overall, despite differences in two key heme metabolism genes, we did not uncover evidence of impaired heme metabolism in *Ahr^-/-* mice that could explain the degree of elevated plasma heme

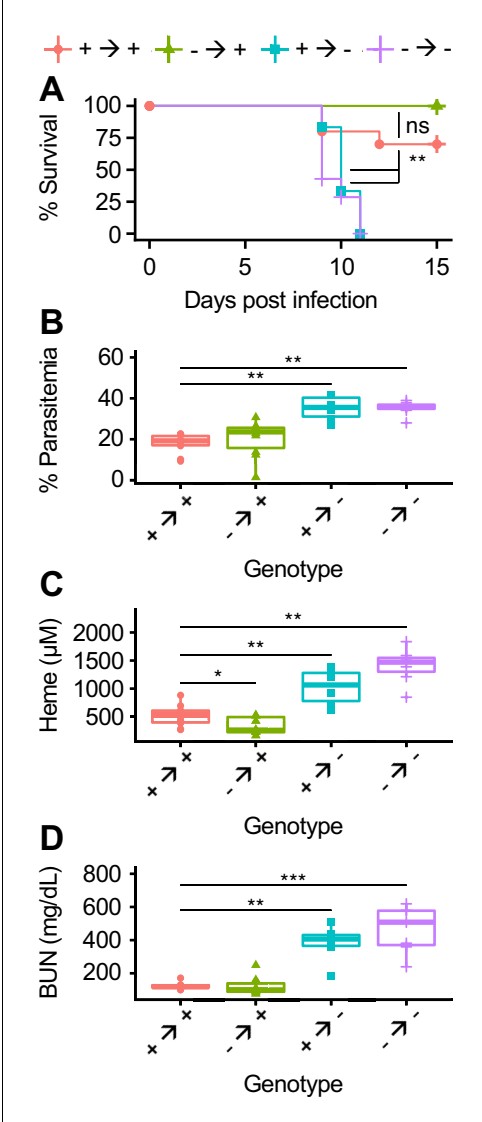

**Figure 5.** AHR is required during *Pc* infection in radioresistant cells. (**A**) Survival, (**B**) parasitemia, and (**C**) total heme (8 DPI), and (**D**) BUN (8 DPI) in *Pc*-infected bone marrow chimeric mice (n = 6–12 per condition). p-Values in A were determined using a log-rank test. p-Values in B-D were determined using a Wilcox test. *p<0.05, **p<0.01, ***p<0.001, ****p<0.0001. Data are representative of two independent experiments.

The online version of this article includes the following source data and figure supplement(s) for figure 5:

**Source data 1.** Source data for *Figure 5*.
**Figure supplement 1.** Neutrophilia inAhr-/-mice during malaria does not cause increased susceptibility.
**Figure supplement 1—source data 1.** Source data for *Figure 5—figure supplement 1*.
**Figure supplement 2.** Increased TNF production inAhr-/-mice during malaria does not cause increased susceptibility.
*Figure 5 continued on next page*

observed during *Pc* infection. We conclude that the loss of AHR leads to increased heme release during malaria, but not impaired heme metabolism or increased heme sensitivity.

## AHR is necessary in radioresistant cells to control parasitemia, plasma heme, and AKI during *Pc* infection

As *Ahr*[-/-] mice mount altered immune responses during *P. berghei* ANKA malaria (*Brant et al., 2014*) and AHR signaling affects immune responses (*Rothhammer and Quintana, 2019*), we first sought to characterize the immune responses of *Ahr*[-/-] mice during infection. *Ahr*[-/-] mice had elevated neutrophil levels in blood during acute sickness (*Figure 5—figure supplement 1A*), in line with other studies linking inappropriate AHR activation to aberrant neutrophil responses in diverse models of inflammation (*Di Meglio et al., 2014*; *Teske et al., 2005*; *Teske et al., 2008*). We found that neutrophil depletion by anti-Ly6G antibody treatment did not affect survival of *Ahr*[-/-] mice (*Figure 5—figure supplement 1B–D*), indicating that AHR-dependent control of neutrophil responses was not responsible for host protection under these conditions. We also observed increased plasma levels of the cytokine TNF in infected *Ahr*[-/-] mice (*Figure 5—figure supplement 2A*), as observed in *P. berghei* ANKA-infected *Ahr*[-/-] mice (*Brant et al., 2014*). We found that TNF neutralization did not alter infection outcome or kidney function in *Ahr*[-/-] mice (*Figure 5—figure supplement 2B–C*). Thus, while we and others observed altered immune responses to malaria in *Ahr*[-/-] mice, we did not identify a link between these differences and *Pc* infection outcome.

To more broadly test the role of AHR in the cellular immune response to malaria, we generated bone marrow chimeric mice. After transplantation and engraftment, chimeric mice had >85% replacement of all measured cell types except for T cells, which averaged 75% donor-derived (*Figure 5—figure supplement 3*). Chimeric mice lacking AHR in radiosensitive cells (- → +) had equivalent survival and parasitemia to wild-type chimeric mice (+ → +) (*Figure 5A–B*). In contrast, mice lacking AHR in radioresistant cells (+ → - and - → -) succumbed to infection and developed higher parasitemia relative to + → + mice (*Figure 5A–B*). We also looked for evidence of heme dysregulation and AKI in susceptible mice. At 8 DPI, + → - and - → - mice had elevated levels of both plasma heme and BUN compared to + → + mice, while - → + mice did not (*Figure 5C–D*). Therefore, during *Pc* infection,

*Figure 5 continued*

**Figure supplement 2—source data 1.** Source data for *Figure 5—figure supplement 2*.

**Figure supplement 3.** Efficiency of bone marrow transplantation measured by flow cytometry on peripheral blood 2 months after transplantation (n = 6–12 per condition).

**Figure supplement 3—source data 1.** Source data for *Figure 5—figure supplement 3*.

AHR in radioresistant cells is necessary to promote survival, limit parasitemia, control plasma heme, and prevent AKI.

## AHR is necessary in *Tek*-expressing cells to control parasitemia, plasma heme, and AKI during *Pc* infection

During *Pc* infection, the sequestration of infected RBCs to endothelial cells in the microvasculature leads to endothelial activation and disrupted barrier integrity (*Miller et al., 2013*). Foreign antigens in the blood interact with endothelial cells, which can promote innate and adaptive immune responses (*Danese et al., 2007*; *Konradt and Hunter, 2018*). AHR is also expressed in endothelial cells, where it regulates blood pressure and vascular development (*Agbor et al., 2011*; *Walisser et al., 2005*). Due to the intimate interactions between parasites and the endothelium, we hypothesized that AHR may be required in endothelial cells during malaria. We generated $Ahr^{fl/fl}::Tek^{cre/+}$ mice, in which AHR is deleted in *Tek*-expressing cells, including endothelial and hematopoietic cells (*Kisanuki et al., 2001*; *Koni et al., 2001*). $Ahr^{fl/fl}::Tek^{cre/+}$ mice were more susceptible to malaria than control $Ahr^{fl/fl}$ mice (*Figure 6A*) and had elevated parasitemia (*Figure 6B*). $Ahr^{fl/fl}::Tek^{cre/+}$ mice also developed higher plasma heme and BUN than control mice (*Figure 6C–D*). These experiments reveal that AHR is required in *Tek*-expressing cells, and previous experiments ruled out a requirement for AHR in radiosensitive cells (*Figure 5*). Therefore, AHR is necessary in *Tek*-expressing, radioresistant cells for survival, control of parasitemia, and limiting AKI during *Pc* infection. Several cell types fit these criteria, including endothelial cells, pericytes, and embryonically derived tissue-resident macrophages (*Kisanuki et al., 2001*; *Koni et al., 2001*; *Teichert et al., 2017*). Our data cannot differentiate between these possibilities.

## Discussion

Altered systemic metabolism is a hallmark of infection. While several studies have dissected the roles of individual metabolic alterations and their effects on infection outcome, the impact of many metabolic changes remains unclear (*Clark et al., 2013*; *Cumnock et al., 2018*; *Dionne et al., 2006*; *Ganeshan et al., 2019*; *Luan et al., 2019*; *Wang et al., 2018*). In this study, we characterized host metabolism during infection by performing metabolomics on mouse plasma during malaria. Like others, we observed extensive reprogramming of host metabolism during infection, with the majority of measured metabolites altered during acute sickness. Mice develop anorexia during acute *Pc* infection (*Cumnock et al., 2018*), and we found altered energy metabolism during acute infection, such as decreases in food-derived compounds and increases in lipid subsets including glycerolipids, glycerophospholipids, and sphingolipids (*Figure 1D*). Compound identification in metabolomics requires known references; therefore, our dataset is biased toward well-studied compounds such as these lipid classes. Nevertheless, we identified nearly 400 metabolites of diverse functions with altered intensity during *Pc* infection.

We observed that each day of infection is marked by a unique combination of immune, metabolic, and pathological events. We propose that therapeutic interventions must be matched to the appropriate stage of infection. For example, antimalarials may be less effective than anemia-targeting interventions for a patient in the latter half of acute infection when parasitemia has dropped significantly. By linking metabolism to the pathology and immune responses occurring at each stage of infection, we developed a rich dataset that allows interrogation of links between metabolism, immune responses, and host physiology during infection.

To this end, we observed that several heme and tryptophan metabolites reached peak intensity in plasma during acute infection. These metabolites can activate AHR, an immunomodulatory transcription factor; in the absence of AHR, mice develop AKI and succumb to infection. While we hypothesized that AHR functions primarily during acute infection, $Ahr^{-/-}$ mice develop increased parasitemia relative to control mice as early as 4 DPI. Metabolites are likely found at different concentrations in plasma and the functionally relevant microenvironment, which may explain this discrepancy.

It is also possible that AHR has distinct roles in early and acute infection, but the effects of increased parasitemia overshadow other AHR-dependent phenotypes. Although the role of kynurenine signaling via AHR is important during certain infection contexts, we observed that survival during malaria under our experimental conditions is kynurenine-independent (*Figure 3—figure supplement 2D*; *Bessede et al., 2014*). Our data suggest that survival requires AHR signaling via one or more of AHR's other endogenous agonists.

While global loss of AHR causes aberrant immune responses, we observed that AHR is dispensable in radiosensitive cells during malaria (*Brant et al., 2014*); instead, AHR in *Tek*-expressing radioresistant cells is required to control parasites and plasma heme, preventing AKI and death (*Figures 5–6*). This phenotype may be dependent on one or more of the multiple cell types that meet these criteria, such as endothelial cells and yolk sac-derived tissue-resident macrophages. Further experimentation is warranted to more precisely identify the relevant cell type(s).

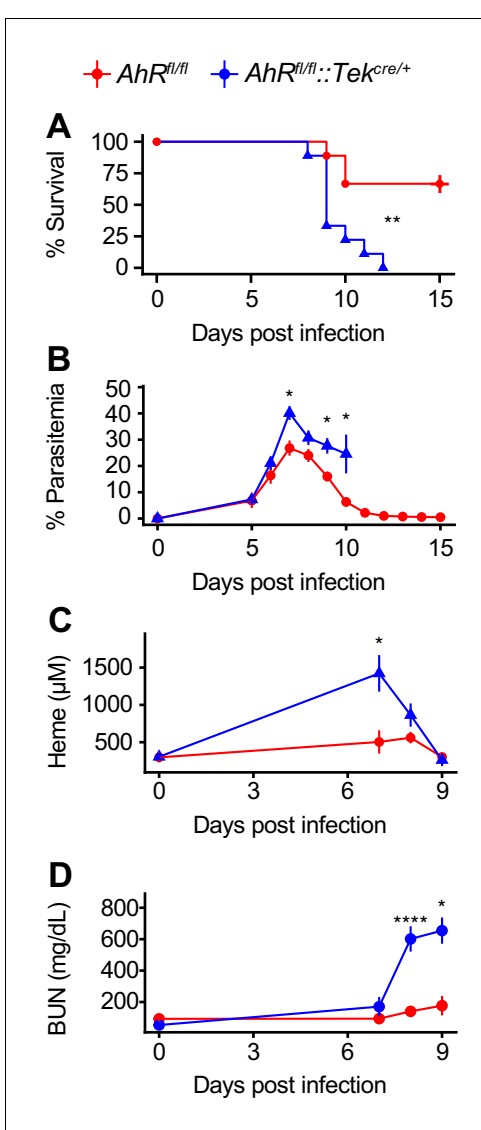

**Figure 6.** AHR is required during *Pc* infection in *Tek*-expressing cells. (A) Survival, (B) parasitemia, (C) plasma heme, and (D) BUN in *Pc*-infected *Ahr^{fl/fl}* and *Ahr^{fl/fl}::Tek^{cre/+}* mice (n = 9 per genotype). p-Values in A were determined using a log-rank test. p-Values in B–D were determined using a two-way ANOVA with FDR correction. *p<0.05, **p<0.01, ***p<0.001, ****p<0.0001. Data are representative of two independent experiments.

The online version of this article includes the following source data for figure 6:

**Source data 1.** Source data for *Figure 6*.

The specific mechanism linking AHR, plasma heme, and AKI is unclear. *Ahr^{-/-}* mice infected with *Pc* or *P. berghei* ANKA show defects in immune responses and parasite control prior to any kidney pathology or defects in heme regulation (*Figures 3* and *4*, *Figure 4—figure supplement 2*; *Brant et al., 2014*). Our data suggest a role for AHR in pathogen recognition and/or innate immune responses in non-hematopoietic cells, two possibilities observed in other contexts (*Di Meglio et al., 2014*; *Moura-Alves et al., 2014*). Thus, a simple model suggests that the increased parasitemia in *Ahr^{-/-}* mice leads to increased hemolysis, in turn causing heme overload and AKI; AHR only functions to control parasite load. This model attributes plasma heme increases in *Ahr^{-/-}* mice solely to increased parasitemia and hemolysis, which may overlook other contributing factors. Nevertheless, if AKI is simply a function of parasitemia, hemolysis, and plasma heme, then increased parasitemia, caused by any mechanism, would also lead to AKI. Treating AKI during malaria would promote disease tolerance rather than resistance to *Plasmodium* (*Ramos et al., 2019*).

Alternatively, we considered a role for AHR in both pathogen control and heme metabolism. This model suggests that *Ahr^{-/-}* mice have impaired heme metabolism. In fact, we observed proper expression of heme metabolism proteins in the liver and kidney of *Ahr^{-/-}* mice (*Figure 4—figure supplement 2C–D*), although failure of infected *Ahr^{-/-}* mice to upregulate *Hrg1* and *Slc40a1* in the liver may merit future study. We also did not specifically evaluate radioresistant *Tek*-expressing cells that contribute to heme metabolism, such as endothelial cells or tissue-resident macrophages (*Balla et al., 1993*; *Petrillo et al., 2018*; *Soares and Hamza, 2016*).

Further undermining this model, $Ahr^{+/-}$ and $Ahr^{-/-}$ mice fare equivalently when treated with phenylhydrazine, a model of sterile heme overload (*Figure 4—figure supplement 3*). While phenylhydrazine treatment recapitulates the acute hemolysis characteristic of malaria, it does not release PAMPs, which may be an important component of the AKI observed in $Ahr^{-/-}$ mice. Overall, our data do not support a role for AHR in heme metabolism.

A third possibility is that AHR functions in kidney-intrinsic protection as well as pathogen control. In *Pc*-infected $Ahr^{-/-}$ mice, impaired kidney function precedes increased plasma heme (*Figure 4B and E*). This suggests that the absence of AHR may initially impair kidney function; elevated plasma heme is a result, not a cause, of kidney injury. A strength of this model is that it links heme metabolism and the production of the AHR ligands biliverdin and bilirubin to AHR activation; it also explains the elevation of plasma heme in *Pc*-infected $Ahr^{-/-}$ mice. Contrary to this model, however, $Ahr^{fl/fl}::$ $Tek^{cre/+}$ mice and chimeric mice with $Ahr^{-/-}$ radioresistant cells both develop AKI only after elevation of plasma heme (*Figure 5*, *Figure 6*, data not shown). Further studies will be crucial to understand the causal relationship between plasma heme and AKI during *Plasmodium* infection, as well as the role of AHR in tissue protection.

As our appreciation grows for the contribution of metabolic adaptation to survival of infection, an important priority is to understand the effects of the many metabolic changes that occur during infection. Recent studies have dissected the contribution of specific metabolic alterations to infection outcome (*Bambouskova et al., 2018*; *Clark et al., 2013*; *Cumnock et al., 2018*; *Dionne et al., 2006*; *Ganeshan et al., 2019*; *Wang et al., 2016*; *Wang et al., 2018*; *Weis et al., 2017*). Our longitudinal study characterized systemic host metabolism during both sickness and recovery. After observing that host-derived AHR ligands increase in concentration during acute infection, we demonstrated that AHR is critical in endothelial cells to limit parasitemia and control tissue damage during malaria, in addition to its better-established roles in immune cells. Overall, these findings suggest that AKI, a common complication of severe malaria, may be caused by heme toxicity, and that therapeutically targeting AHR and/or heme metabolism, rather than parasites, may improve malaria outcomes without driving drug resistance in *Plasmodium*. We also expect that other metabolites altered during infection have similarly important biological functions, and these relationships may be a rich area for future study.

# Materials and methods

## Key resources table

| Reagent type (species) or resource | Designation | Source or reference | Identifiers | Additional information |
|---|---|---|---|---|
| Antibody | TruStain FcX Antibody | Biolegend | BioLegend Cat# 101319, RRID:AB_1574973 | |
| Antibody | Mouse monoclonal CD71 PerCP/Cy5.5 (clone RI7217) | Biolegend | BioLegend Cat# 113815, RRID:AB_2565481 | |
| Antibody | Mouse monoclonal TER-119 PE/Cy7 (clone TER-119) | Biolegend | BioLegend Cat# 116221, RRID:AB_2137789 | |
| Antibody | Mouse monoclonal TCR γ/δ PE (clone UC7-13D5) | Biolegend | BioLegend Cat# 107507, RRID:AB_345265 | |
| Antibody | Mouse monoclonal CD19 BV 785 (clone 6D5) | Biolegend | BioLegend Cat# 115543, RRID:AB_11218994 | |
| Antibody | Mouse monoclonal CD3 BV 650 (clone 17A2) | Biolegend | BioLegend Cat# 100229, RRID:AB_11204249 | |

*Continued on next page*

*Continued*

| Reagent type (species) or resource | Designation | Source or reference | Identifiers | Additional information |
|---|---|---|---|---|
| Antibody | Mouse monoclonal CD8a BV510 (clone 53–6.7) | Biolegend | BioLegend Cat# 100751, RRID:AB_2561389 | |
| Antibody | Mouse monoclonal Ly-6G BV 421 (clone 1A8) | Biolegend | BioLegend Cat# 127628, RRID:AB_2562567 | |
| Antibody | Mouse monoclonal CD4 Alexa Fluor 700 (clone RM4-5) | Biolegend | BioLegend Cat# 100536, RRID:AB_493701 | |
| Antibody | Mouse monoclonal Ly-6C Alexa Fluor 647 (clone HK1.4) | Biolegend | BioLegend Cat# 128010, RRID:AB_1236550 | |
| Antibody | Mouse monoclonal CD335 FITC (clone 29A1.4) | Biolegend | BioLegend Cat# 137606, RRID:AB_2298210 | |
| Antibody | Mouse monoclonal CD11b APC-eFluor 780 (clone M1/70) | Thermo Fisher Scientific | Thermo Fisher Scientific Cat# 47-0112-82, RRID:AB_1603193 | |
| Antibody | Mouse monoclonal CD41 BUV395 (clone MWReg30) | BD Biosciences | BD Biosciences Cat# 565980, RRID:AB_2739432 | |
| Antibody | Human monoclonal Heme Oxygenase 1 (clone EP1391Y) | Abcam | Abcam Cat# ab52947, RRID:AB_880536 | (1:2000 dilution) |
| Antibody | Mouse monoclonal Ferritin Heavy Chain (clone EPR18878) | Abcam | Abcam Cat# ab183781 | (1:2000 dilution) |
| Antibody | Mouse monoclonal DMT1 antibody | Abcam | Abcam Cat# ab55735, RRID:AB_2239227 | (1:400 dilution) |
| Antibody | Mouse polyclonal HCP1/PCFT antibody | Abcam | Abcam Cat# ab25134, RRID:AB_2270597 | (1:1000 dilution) |
| Antibody | Mouse monoclonal β-Actin (clone AC-15) | Sigma Aldrich | Sigma-Aldrich Cat# A1978, RRID:AB_476692 | (1:2000 dilution) |
| Antibody | Rabbit polyclonal IgG HRP | GE Healthcare | GE Healthcare Cat# GENA934, RRID:AB_2722659 | (1:10,000 dilution) |
| Antibody | Mouse polyclonal IgG HRP | Bio-Rad | Bio-Rad Cat# 170–6516, RRID:AB_11125547 | (1:3000 dilution) |
| Antibody | Mouse polyclonal CD45.2 PerCP-Cy5.5 (clone 104) | Thermo Fisher Scientific | Thermo Fisher Scientific Cat# 45-0454-82, RRID:AB_953590 | |
| Antibody | Mouse monoclonal NK-1.1 FITC (clone PK136) | Biolegend | BioLegend Cat# 108705, RRID:AB_313392 | |
| Antibody | Mouse monoclonal CD11c PE-Cy7 (clone N418) | Thermo Fisher Scientific | Thermo Fisher Scientific Cat# 25-0114-82, RRID:AB_469590 | |

*Continued on next page*

*Continued*

| Reagent type (species) or resource | Designation | Source or reference | Identifiers | Additional information |
|---|---|---|---|---|
| Antibody | Mouse monoclonal CD45.1 PE (clone A20) | Biolegend | BioLegend Cat# 110707, RRID:AB_313496 | |
| Antibody | Mouse monoclonal Ly6G (clone 1A8) | Bio X Cell | Bio X Cell Cat# BE0075-1, RRID:AB_1107721 | |
| Antibody | Rat monoclonal IgG2a, (clone 2A3) | Bio X Cell | Bio X Cell Cat# BE0089, RRID:AB_1107769 | |
| Antibody | Mouse monoclonal TNFα (clone XT3.11) | Bio X Cell | Bio X Cell Cat# BE0058, RRID:AB_1107764 | |
| Antibody | Rat monoclonal IgG1 (clone HRPN) | Bio X Cell | Bio X Cell Cat# BE0088, RRID:AB_1107775 | |
| Chemical compound, drug | Formic acid, reagent grade, ≥95% | Sigma Aldrich | Cat. F0507-100ML | |
| Chemical compound, drug | Hemin | Sigma Aldrich | Cat. H9039 | |
| Chemical compound, drug | Phenylhydrazine, 97% | Sigma Aldrich | Cat. P26252-100G | |
| Chemical compound, drug | L-Tryptophan | Sigma Aldrich | Cat. T0254 | |
| Chemical compound, drug | Tryptophan-d5 | Medical Isotopes | Cat. D34384 | |
| Chemical compound, drug | L-Kynurenine | Sigma Aldrich | Cat. K8625 | |
| Chemical compound, drug | L-Kynurenine-d4 | Medical Isotopes | Cat. D37563 | |
| Chemical compound, drug | 3-Hydroxy-DL-kynurenine | Sigma Aldrich | Cat. H1771 | |
| Chemical compound, drug | 3-Hydroxy-DL-kynurenine-d3 | Medical Isotopes | Cat. D34384 | |
| Chemical compound, drug | Kynurenic acid | Sigma Aldrich | Cat. 67667 | |
| Chemical compound, drug | Kynurenic acid-3,5,6,7,8-d5 | Sigma Aldrich | Cat. 793477 | |
| Chemical compound, drug | 2,3-Pyridinedicarboxylic acid | Sigma Aldrich | Cat. P63204 | |
| Chemical compound, drug | Quinolinic Acid-d3 | Medical Isotopes | Cat. D18880 | |
| Commercial assay or kit | Bilirubin Assay Kit | Sigma Aldrich | Cat. MAK126-1KT | |

*Continued on next page*

*Continued*

| Reagent type (species) or resource | Designation | Source or reference | Identifiers | Additional information |
|---|---|---|---|---|
| Commercial assay or kit | ALT Activity Assay | Sigma Aldrich | Cat. MAK052-1KT | |
| Commercial assay or kit | Bioassay Systems Urea Assay Kit | Fisher Scientific | Cat. 50-107-8333 | |
| Commercial assay or kit | TruSeq RNA Library Prep Kit v2 | Illumina | Cat. RS-122–2001 | |
| Commercial assay or kit | TNF alpha Mouse ELISA Kit | Thermo Fisher Scientific | Thermo Fisher Scientific Cat# BMS607/3, RRID:AB_2575663 | |
| Genetic reagent (*Mus musculus*) | $Ahr^{-/-}$ mice | Taconic | Cat. 9166, RRID:IMSR_TAC:9166 | |
| Genetic reagent (*M. musculus*) | $Ahr^{+/+}$ mice | Taconic | Cat. B6, RRID:IMSR_TAC:b6 | |
| Genetic reagent (*M. musculus*) | $Ido1^{-/-}$ mice | Jax | Cat. 005867, RRID:IMSR_JAX:005867 | |
| Genetic reagent (*M. musculus*) | C57BL/6J mice | Jax | Cat. 000664, RRID:IMSR_JAX:000664 | |
| Genetic reagent (*M. musculus*) | $AhR^{fl/fl}$ mice | Jax | Cat. 006203, RRID:IMSR_JAX:006203 | |
| Genetic reagent (*M. musculus*) | $Tek^{cre/+}$ mice | Jax | Cat. 004128, RRID:IMSR_JAX:004128 | |
| Genetic reagent (*M. musculus*) | CD45.1 mice | Taconic | Cat. 4007, RRID:IMSR_TAC:4007 | |
| Genetic reagent (*M. musculus*) | C57BL/6NCrl mice | Charles River | Cat. 027, RRID:IMSR_CRL:027 | |
| Genetic reagent (*Plasmodium chabaudi*) | Plasmodium chabaudi AJ | Malaria Research and Reference Reagent Resource Center | Cat. MRA-756 | |
| Commercial assay or kit | Bilirubin Assay Kit | Sigma Aldrich | Cat. MAK126-1KT | |
| Commercial assay or kit | RNeasy Mini kit | Qiagen | Cat. 74104 | |
| Commercial assay or kit | Rnase-Free Dnase | Qiagen | Cat. 79256 | |
| Commercial assay or kit | SuperScript III First-Strand synthesis system | Fisher Scientific | Cat. 18-080-051 | |
| Commercial assay or kit | FastStart Universal SYBR Green Master (Rox) | Sigma Aldrich | Cat. 04913850001 | |
| Software, algorithm | FlowJo 10.0.8r1 | Tree Star | https://www.flowjo.com/solutions/flowjo | |

*Continued on next page*

Continued

| Reagent type (species) or resource | Designation | Source or reference | Identifiers | Additional information |
|---|---|---|---|---|
| Software, algorithm | R v3.4.0 | R | https://www.r-project.org/ | |
| Software, algorithm | ggplot2 v3.1.0 | ggplot2 | https://github.com/tidyverse/ggplot2 | |

## Mice

Mice were housed in the Stanford Research Animal Facility according to Stanford University guidelines. The Stanford Administrative Panel on Laboratory Animal Care approved all mouse experiments. Female mice were used unless otherwise indicated. C57BL/6N mice were from Charles River Laboratories were used for the cross-sectional metabolomics experiment. $Ahr^{-/-}$ mice were originally obtained from Taconic (C57BL/6-$Ahr^{tm1.2Arte}$; 9166) and crossed with $Ahr^{+/+}$ mice (C57BL/6NTac; B6-F) to generate $Ahr^{+/-}$ mice. $Ahr^{+/-}$ x $Ahr^{+/-}$ crosses produced $Ahr^{+/+}$, $Ahr^{+/-}$, and $Ahr^{-/-}$ mice. For higher yields of $Ahr^{-/-}$ mice, $Ahr^{+/-}$ females were crossed to $Ahr^{-/-}$ males. Female $Ido1^{-/-}$ and $Ido1^{+/+}$ mice were obtained from Jax (B6.129-$Ido1^{tm1Alm}$/J; 005867 and C57BL/6J; 000664).

$Ahr^{fl/fl}::Tek^{cre/+}$ mice were generated by crossing C57BL/6J $Ahr^{fl/fl}$ mice ($Ahr^{tm3.1Bra}$/J; 006203) (*Walisser et al., 2005*) and C57BL/6J $Tek^{cre/+}$ mice (B6.Cg-Tg(Tek-cre)12Flv/J; 004128) (*Koni et al., 2001*). Only male $Tek^{cre/+}$ mice were used for breeding.

## Infections

Age-matched littermates were separated by genotype and infected at 8–12 weeks old unless otherwise indicated. All *Plasmodium chabaudi* AJ strain parasites were obtained from the Malaria Research and Reference Reagent Resource Center (MR4) and were tested for contaminating pathogens prior to use. Female passage mice were given intraperitoneal (IP) injections of frozen stocks of *Pc*-infected RBCs (iRBCs). To measure parasitemia, 2 µl tail blood was collected via tail nicking of restrained mice using sterilized surgical scissors. A thin blood smear was prepared on microscope slides (Globe Scientific 1324), fixed in methanol (Fisher Scientific A454SK-4), and stained with Giemsa (Thermo Fisher Scientific 10092013), and the percentage of iRBCs was counted at 100x magnification. An additional 2 µl blood was diluted in 1 ml of Hanks' Balanced Salt Solution (Fisher Scientific 14185052) to count the number of RBCs/µl blood. Absolute counts were obtained on an Accuri C6 flow cytometer using forward and side scatter. Once parasitemia reached 10–20% (7–9 days), $10^5$ freshly obtained iRBCs diluted in sterile Krebs saline with glucose (KSG; 0.1 M NaCL, 4.6 mM KCl, 1.2 mM MgSO$_4$*7 $_H$20, 0.2% glucose (w/v), pH 7.4) were IP injected into experimental animals. Uninfected control animals were injected with KSG alone.

## Cross-sectional infection

Age-matched C57BL/6N mice were purchased from Charles River Laboratories and infected as one cohort. Five infected mice were sacrificed every day post-infection, five uninfected mice were sacrificed on 0 dpi, and two uninfected mice were sacrificed on days 5, 8, 10, 12, 15, 19, and 25 (a total of 19 uninfected samples across infection). Sample collection was performed as follows: After the collection of 4 µl of blood were collected from each animal (2 µl for thin blood smears and 2 µl for RBC counts), animals were euthanized by $CO_2$ inhalation per Stanford University guidelines. Blood was then collected via cardiac puncture into 100 µl of 0.5 M EDTA, pH 8.0. Some of this blood was used for flow cytometry analysis (~5–12 µl), and the remainder was spun for 5 min at 1000 x g to collect plasma for ALT quantification, Luminex, and metabolomics analysis. All samples were stored at −80°C for later processing.

## Metabolomics and analysis

100 µl of plasma were sent to Metabolon (http://www.metabolon.com), which performed a combination of gas and liquid chromatography techniques combined with mass spectrometry (GC/LC-MS). A table of 587 detected metabolites was returned with the raw area count, which were normalized by

dilution and rescaled to set the median equal to 1 ('scaled intensity'). Median fold change (MFC) for each metabolite at each timepoint was calculated relative to the median value of uninfected day 0 samples. Next, the greatest magnitude MFC was identified for each metabolite, and metabolites with absolute value MFC <2 were removed from further analysis. Lastly, 364 significantly changed metabolites were identified by comparing the time point with the greatest magnitude MFC to the day 0 time point (adjusted p-value<0.05 by t-test with FDR correction).

## Luminex

This assay was performed in the Human Immune Monitoring Center at Stanford University. Mouse 38-plex kits were purchased from eBiosciences/Affymetrix and used according to the manufacturer's recommendations with modifications as described below. Briefly, beads were added to a 96-well plate and washed in a Biotek ELx405 washer. 60 µl of plasma per sample were submitted for processing. Samples were added to the plate containing the mixed antibody-linked beads and incubated at room temperature for one hour followed by overnight incubation at 4°C with shaking. Cold and room temperature incubation steps were performed on an orbital shaker at 500–600 rpm. Following the overnight incubation, plates were washed as above and then a biotinylated detection antibody was added for 75 min at room temperature with shaking. Plates were washed as above and streptavidin-PE was added. After incubation for 30 min at room temperature a wash was performed as above and reading buffer was added to the wells. Each sample was measured as singletons. Plates were read using a Luminex 200 instrument with a lower bound of 50 beads per sample per cytokine. Custom assay control beads by Radix Biosolutions were added to each well. Each cytokine was normalized to its median value on each plate. Significance was calculated by comparing each infected timepoint to values from uninfected mice across infection.

## Flow cytometry

In experiments assessing general immune cell classes in the blood, approximately 10 million cells were plated in FACS buffer (PBS, 0.2% fetal bovine serum (Sigma), 5 mM EDTA). Prior to staining, the cells were incubated in TruStain FcX amntibody (Biolegend) for at least 5 min at 4°C. A cocktail containing the Live/Dead Fixable Blue stain (Fisher L34962) and antibodies against the following antigens was added to the blocked cells: CD71 PerCP-Cy5.5 (clone RI7217), TER-119 PE-Cy7 (TER-119), TCRγδ PE (UC7-13D5), CD19 Brilliant Violet (BV) 785 (6D5), CD3 BV650 (17A2), CD8 BV510 (53–6.7), Ly6G BV421 (1A8), CD4 Alexa Fluor 700 (GK1.5), Ly6C Alexa Fluor 647 (HK1.4), CD335 FITC (29A1.4) (all from Biolegend); CD11b Alexa 780 (M1/70, eBioscience); CD41 BUV395 (MWReg30, BD Biosciences). All stains were performed for 12–15 min at 4°C. 5 µl of CountBright counting beads (Invitrogen) were added to each samples such that absolute counts per µl of blood could be back calculated. Data were acquired on an LSR Fortessa (BD Biosciences) and analyzed using FlowJo 10.0.8r1 (Tree Star). Significance was calculated by comparing each infected timepoint to values from uninfected mice across infection.

## Mass spectrometry

The analytes were tryptophan (TRP), kynurenine (KYN), 3-hydroxykynurenine (3HK), kynurenic acid (KA), and quinolinic acid (QA) and the internal standards were tryptophan-d5 (TRP-d5), kynurenine-d4 (KYN-d4), 3-hydroxykynurenine-d3 (3HK-d3), kynurenic acid-d5 (KA-d5), quinolinic acid-d3 (QA-d3).

Individual analyte primary stock solutions (10 mM) were prepared in DMSO (KYN, KA); 0.1% formic acid in water (TRP, QA); or 0.45 N HCl in 0.1% formic acid water (3HK). Intermediate stock solution consisting of five analytes: TRP; KYN; KA; 3HK; QA, was prepared from individual primary stock solutions. This intermediate stock solution was serially diluted with 0.1% formic acid/0.02% L-ascorbic acid in water to obtain a series of standard working solutions which were used to generate the calibration curve. Standard working solutions were prepared freshly for sample analysis. Calibration curves were prepared by spiking 10 µl of each of the standard working solutions into 50 µl of PBS/0.02% ascorbic acid followed by addition of 10 µl internal standard solution consisting of five analytes (25 µM TRP-d5; 5 µM KYN-d4, 3HK-d3, KA-d5; 7.5 µM QA-d3). Because of interference due to endogenous tryptophan and metabolites, calibration curves were not prepared in the same matrix (plasma) as the study samples. Blank charcoal stripped plasma still contained quantifiable amounts

of tryptophan and metabolites. A calibration curve was prepared fresh with each set of samples. Calibration curve ranges: for KYN and KA, 1 nM to 10 µM; for 3HK, 2.5 nM to 10 µM; for QA, 5 nM to 10 µM; for TRP, 10 nM to 200 µM.

Fifty µl aliquots of plasma were used for analysis. 10 µl internal standard solution was added to 50 µl plasma aliquot followed by vortexing. 200 µl ice cold solution of methanol/1% acetic acid/0.02% L-ascorbic acid was added to the sample, followed by vortexing, then centrifugation. Supernatant was transferred to a new vial, evaporated to dryness under nitrogen, reconstituted in 50 µl 0.1% formic acid/0.02% ascorbic acid in water and analyzed by LC-MS/MS. L-ascorbic acid and evaporation under nitrogen ($N_2$) gas were used to prevent oxidation. For QA determination, standard samples and plasma samples were diluted 10-fold with 0.1% formic acid/0.02% ascorbic acid in water and 10 µl injected to LC-MS/MS.

All analyses were carried out by positive electrospray LC-MS/MS using an LC-20AD$_{XR}$ Prominence liquid chromatograph and 8030 triple quadrupole mass spectrometer (Shimadzu). HPLC conditions: Atlantis T3 2.1 × 100 mm, 3 µm particle size column was operated at 45°C at a flow rate of 0.25 mL/min. Mobile phases consisted of A: 0.2% formic in water and B: 0.2% formic acid in acetonitrile. Elution profile: initial hold at 0% B for 1 min, followed by a gradient of 0–30% in 6 min, then 30–95% in 2 min, equilibrating back to 0% B; total run time was 13 min. Injection volume was 10 µl.

Selected reaction monitoring (SRM) was used for quantification. Analyte mass transitions were as follows: TRP: m/z 205.0 → m/z 146.0 (quantifier) and m/z 205.0 → m/z 118.0 (qualifier); KYN: m/z 209.0 → m/z 94.1 (quantifier) and m/z 209.1 → m/z 146.0 (qualifier); 3HK: m/z 225.0 → m/z 208.1 (quantifier) and m/z 225.0 → m/z 110.1 (qualifier); KA: m/z 189.9 → m/z 89.1 (quantifier) and m/z 189.9 → m/z 116.1 (qualifier); QA: m/z 167.9 → m/z 78.1 (quantifier) and m/z 167.9 → m/z 105.9 (qualifier). For internal standards: TRP-d5: m/z 210.0 → m/z 150.1; KYN-d4: m/z 213.0 → m/z 98.2; 3HK-d3: m/z 228.0 → m/z 111.1; KA-d5: m/z 195.0 → m/z 121.1; QA-d3: m/z 170.9 → m/z 81.1. Dwell time was 20–30 ms.

Quantitative analysis was done with LabSolutions LCMS software (Shimadzu) using an internal standard approach. Calibration curves were linear (R > 0.99) over the concentration range using a weighting factor of $1/X^2$ where X is the concentration. The back-calculated standard concentrations were ±15% from nominal values, and ±20% at the lower limit of quantitation (LLOQ).

## Longitudinal infection monitoring

Sampling was performed as described previously (*Cumnock et al., 2018*; *Torres et al., 2016*) between 7AM-12PM. Temperature was measured by rectal probe (Physitemp Instruments Inc BAT-12 and World Precision Instruments RET-3) and was recorded daily with weight. Mice were restrained and approximately 16 µl of tail blood was collected as described above. Thin blood smears were generated using 2 µl blood, and parasitemia and RBCs/µl blood were measured as described above. Parasite density was calculated by multiplying the percent parasitemia from the blood smears by the daily RBC counts. An additional 12 µl blood was collected for other purposes. Tails were bled gently to prevent hemolysis from pressure. Age-matched mice were sampled as described except that only 4 µl blood was collected (2 µl for parasitemia, 2 µl for measuring anemia).

## Histology

Mice were euthanized and portions of liver, kidney, lung, and spleen were harvested for histology, fixed in 10% formalin (VWR 50-420-850), routinely processed, embedded in paraffin, sectioned, and stained with hematoxylin and eosin and Perls Prussian blue as indicated. Blinded slides were evaluated by a veterinary pathologist using an Olympus BX43 upright brightfield microscope. Photomicrographs were captured using an Olympus DP27 camera and the Olympus cellSens software.

## RNA isolation and qRT-PCR

Mice were euthanized at the indicated timepoints. When indicated, perfusion was performed by cutting the vena cava and slowly introducing 10 ml of cold PBS into circulation via the heart. Tissues were dissected, snap-frozen in liquid nitrogen, and transferred to −80°C. RNA was isolated from thawed tissue (30–50 mg) using the RNeasy Mini kit (Qiagen 74104) and treated with DNAse (Qiagen 79256). cDNA was synthesized from 1 µg of RNA using SuperScript III First-Strand synthesis

system (Fisher Scientific 18-080-051). Transcripts were amplified using FastStart Universal SYBR Green Master (Rox; Millipore Sigma 04913850001) and gene-specific primers (*Table 1*).

## Western blotting

Dissected tissues were snap-frozen in liquid nitrogen and transferred to −80˚C. Approximately 100 mg of tissue was homogenized in RIPA buffer (50 mM Tris pH 8.0, 150 mM NaCl, 0.1% SDS (w/v), 0.5% sodium deoxycholate (w/v), 1% Triton X-100 (v/v)) with 1X protease inhibitors (Millipore Sigma 11836170001). Total protein content was measured by Bradford (Fisher Scientific PI23200) and 25 µg protein was diluted in 1X Laemmli Sample Buffer (Bio-Rad 1610747) containing β-mercaptoethanol. Samples were incubated at 95˚C for 5 min, separated by SDS-PAGE on 4–12% polyacrylamide gels (Thermo Fisher Scientific NP0323BOX) and transferred to PVDF membranes (Bio-Rad 1704156) using the TransBlot Turbo System (Bio-Rad). Membranes were blocked in 5% nonfat milk dissolved 1X PBS (1 g NACl,. 2 g KCl, 1.44 g $Na_2HPO_4$ dibasic,. 24 g $KH_2PO_4$ monobasic dissolved in 1 L water, adjusted pH to 7.4 and autoclaved) containing 0.1% Tween-20 (Millipore Sigma P1379) at room temperature for 1 hr. Membranes were incubated with primary antibodies against HO-1 (Abcam ab52947, 1:2000), FTH (Abcam ab183781, 1:2000), DMT1 (Abcam ab55735, 1:400), HCP1 (Abcam ab25134, 1:1000), and β-Actin (Sigma A1978, 1:2000) overnight at 4˚C, washed with PBST, and incubated with anti-rabbit IgG-HRP (Sigma GENA934, 1:10,000) or anti-mouse IgG-HRP (Bio-Rad 1706516, 1:3000) for 1 hr at room temperature. Membranes were again washed with PBST and HRP signal was detected using SuperSignal West Femto Chemiluminescent Substrate (Fisher Scientific PI34095) on a ChemiDoc imager (Bio-Rad).

## Quantification of tissue injury markers and plasma compounds

Blood from cardiac punctures or tail bleeds were processed into plasma as described above. In the cross-sectional infection experiment, ALT was measured on a Dimension Xpand analyzer (Siemens). A medical technologist performed all testing and reviewed all data. For all other experiments, bilirubin, ALT, and BUN were measured using kits (bilirubin by Millipore Sigma MAK126-1KT, ALT by Millipore Sigma MAK052-1KT, and BUN by Fisher Scientific 50-107-8333). Plasma samples for bilirubin were collected in the dark and measured within 5 hr to minimize UV degradation (*Rehak et al., 2008*). Heme in plasma and urine were measured as described previously (*Ramos et al., 2019*). Plasma and urine were diluted between 1:1000 and 1:25 in water. 150 µl formic acid (Millipore Sigma F0507-100ML) was added and absorbance was measured at 405 nm. Urine samples were also measured at 355 nm and background absorbance was corrected using the formula $\lambda_{405}nm = \lambda_{405}nm \times (\lambda_{405}nm/\lambda_{355}nm)$. Absorbance was compared to a standard curve of hemin (Millipore Sigma H9039-

**Table 1.** qRT-PCR primers used in this study.

| Gene | Forward primer | Reverse primer | Source |
|------|----------------|----------------|--------|
| Hp | GCTATGTGGAGCACTTGGTTC | CACCCATTGCTTCTCGTCGTT | PrimerBank 8850219a1 |
| Hpx | AGCAGTGGCGCTAAATATCCT | CCATTTTCAACTTCGGCAACTC | PrimerBank 23956086a1 |
| Hrg1 | GACGGTGGTCTACCGACAAC | TCCTCCAGTAATCCTGCATGTA | PrimerBank 13385856a1 |
| Hmbs | AAAGTTCCCCAACCTGGAAT | CCAGGACAATGGCACTGAAT | |
| Hmox1 | AAGGAGGTACACATCCAAGCCGAG | GATATGGTACAAGGAAGCCATCACCAG | *Ramos et al., 2019* |
| Fth | CCATCAACCGCCAGATCAAC | GCCACATCATCTCGGTCAAA | *Ramos et al., 2019* |
| Mfsd7b | TCTTCAGCCTTTACTCGCTGG | GAAGTCCTCGAACACGTTGCT | PrimerBank 124486924 c1 |
| Mfsd7c | GGAGAAAGCGATTAGAGAAGGC | CTGATGGCTGCATTTCACAGT | PrimerBank 26340226a1 |
| Slc40a1 | TGCCTTAGTTGTCCTTTGGG | GTGGAGAGAGAGTGGCCAAG | *Ramos et al., 2019* |
| Tfrc | GTTTCTGCCAGCCCCTTATTAT | GCAAGGAAAGGATATGCAGCA | PrimerBank 11596855a1 |
| Msp1 | ACTGAAGCAACAACACCAGC | GTTGTTGATGCACTTGCGGGTTC | *Cheesman et al., 2006* |
| Havcr1 | TGGTTGCCTTCCGTGTCTCT | TCAGCTCGGGAATGCACAA | *Kulkarni et al., 2014* |
| Lcn2 | TGGCCCTGAGTGTCATGTG | CTCTTGTAGCTCATAGATGGTGC | PrimerBank 1019908a1 |
| Arbp0 | CTTTGGGCATCACCACGAA | GCTGGCTCCCACCTTGTCT | *Ramos et al., 2019* |

1G) at 0,. 5, 1, 5, 10, and 20 uM. Day 0 samples were excluded from heme analyses if visual inspection revealed hemolysis caused by the bleeding process; this did not occur on subsequent bleeding. Values from cardiac puncture blood were corrected for the percentage of EDTA in the total volume of the cardiac puncture.

## Bone marrow chimeras

CD45.1 $Ahr^{+/+}$ (B6.SJL-$Ptprc^a$/BoyAiTac; 4007 F), CD45.2 $Ahr^{+/+}$ (C57BL/6NTac; B6-F), and CD45.2 $Ahr^{-/-}$ mice were lethally irradiated (2 × 6 Gy, 6 hr apart) at 5–7 weeks of age. Bone marrow from donor CD45.1 $Ahr^{+/+}$ and CD45.2 $Ahr^{-/-}$ mice was delivered by tail vein injection 1 hr after the second radiation dose. Mice were maintained for 2 weeks on autoclaved food and water containing 2 mg/ml neomycin sulfate (VWR 89149–866) and 1000 U/ml polymyxin B (Millipore Sigma P4932-5MU). Bone marrow engraftment was assessed 8 weeks after transplantation by processing 10 µl of tail vein blood as described above and staining with Live/Dead Fixable Blue stain (Fisher L34962) and the following antibodies: CD45.2 PerCP-Cy5.5 (Fisher/Invitrogen 45-0454-80), NK-1.1 FITC (Biolegend 108706), CD11c PE-Cy7 (Fisher/Invitrogen 25-0114-82), CD45.1 PE (Biolegend 110707), CD19 BV 785 (Biolegend 115543), CD3 BV 650 (Biolegend 100229), CD8a BV 510 (Biolegend). Mice were infected 9 weeks after transplantation. Sampling was performed as described in Longitudinal infection monitoring with the following modification: 12 µl of blood for BUN and heme quantification was collected on day 0, and 7–9 only. Flow panel used for validation.

## Phenylhydrazine treatment

Phenylhydrazine (Sigma Aldrich P26252-100G) was dissolved in sterile PBS immediately before treatment. Mice were I.P. injected with 0.1 mg/g phenylhydrazine in 100 µl.

## RNA-seq

Liver RNA was purified using TRIzol (Fisher 15596026). cDNA libraries were prepared using a TruSeq RNA Library Prep Kit v2 (Illumina RS-122–2001) with 500 ng RNA as input. A HiSeq 4000 (Illumina) was used for sequencing, with a paired-end sequencing length of 75 bp. Sequencing data can be accessed at GSE 150268.

## Neutrophil depletion

Mice were IP injected with 250 µg of either anti-Ly6G clone 1A8 (Bio X Cell BE0075-1) or IgG2a isotype control (Bio X Cell BE0089) in 100 µl of sterile PBS on 5, 6, and 7 DPI. Each day, approximately 16 µl of tail blood was collected for assorted analyses, including flow cytometry as described above. Because treatment with Ly6G interferes with detection of neutrophils, we defined neutrophils as $CD11b^{hi}Ly6C^{int}Ly6G^+$ using a gating strategy as described previously (*Shi et al., 2011*).

## TNF neutralization

Plasma TNF was measured by ELISA (Fisher BMS607-3). To neutralize TNF, mice were IP injected with 500 µg anti-TNF clone XT3.11 (Bio X Cell BP0058) or IgG1 isotype control (Bio X Cell BE0088) in 100 µl of sterile PBS on 7 DPI.

## Acknowledgements

KC is the recipient of an NSF GRFP fellowship. NMD is the recipient of an NSF GRFP (DGE-1656518). This work was supported by NIH grants 5DP1AT007753 and 1R21AI145365 (DSS), the DARPA contract W911NF-16–0052 (DSS), and an NIH training grant (T32 AI007328, MML and NMD). Further support comes from NIH P30 CA124435 utilizing the Stanford Cancer Institute Proteomics/Mass Spectrometry Shared Resource. We thank the Stanford Animal Histology Services for help with preparation of histologic specimens, and the Stanford Animal Diagnostic Lab for help with analyzing samples. We also thank Drs. Juliana Idoyaga and Amanda Jacobson for discussions and critical reading of the manuscript.

## Additional information

### Funding

| Funder | Grant reference number | Author |
|---|---|---|
| National Science Foundation | DGE-1656518 | Nicole M Davis |
| National Science Foundation | | Katherine Cumnock |
| National Institutes of Health | 5DP1AT007753 | David Schneider |
| National Institutes of Health | 1R21AI145365 | David Schneider |
| Defense Advanced Research Projects Agency | W911NF-16-0052 | David Schneider |
| National Institutes of Health | T32 AI007328 | Michelle M Lissner Nicole M Davis |

The funders had no role in study design, data collection and interpretation, or the decision to submit the work for publication.

### Author contributions

Michelle M Lissner, Conceptualization, Resources, Formal analysis, Investigation, Visualization, Writing - original draft, Writing - review and editing; Katherine Cumnock, Conceptualization, Resources, Investigation, Writing - review and editing; Nicole M Davis, Resources, Formal analysis, Investigation, Writing - review and editing; José G Vilches-Moure, Priyanka Basak, Daniel J Navarrete, Jessica A Allen, Investigation, Writing - review and editing; David Schneider, Conceptualization, Supervision, Funding acquisition, Project administration, Writing - review and editing

### Author ORCIDs

Michelle M Lissner (iD) https://orcid.org/0000-0002-3854-2919
Priyanka Basak (iD) http://orcid.org/0000-0002-8550-7892
David Schneider (iD) https://orcid.org/0000-0002-2391-9963

### Ethics

Animal experimentation: Experiments involving animals were performed in accordance with NIH guidelines, the Animal Welfare Act, and US federal law. All animal experiments were approved by the Stanford University Administrative Panel on Laboratory Animal Care (APLAC) and overseen by the Institutional Animal Care and Use Committee (IACUC) under Protocol ID 30923. Animals were housed in a centralized research animal facility accredited by the Association of Assessment and Accreditation of Laboratory Animal Care (AAALAC) International.

### Decision letter and Author response

Decision letter https://doi.org/10.7554/eLife.60165.sa1
Author response https://doi.org/10.7554/eLife.60165.sa2

## Additional files

### Supplementary files

• Source data 1. Raw metabolomics data from *Figure 1*.

• Transparent reporting form

### Data availability

Sequencing data have been deposited in GEO under accession code GSE150268. Untargeted metabolomics data are included in the manuscript and supporting files.

The following dataset was generated:

| Author(s) | Year | Dataset title | Dataset URL | Database and Identifier |
|---|---|---|---|---|
| Lissner ML, Schneider DS | 2020 | Liver transcriptomics during malaria | https://www.ncbi.nlm.nih.gov/geo/query/acc.cgi?acc=GSE150268 | NCBI Gene Expression Omnibus, GSE150268 |

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
