## [Decision Letter]

**Acceptance summary:**

This is a thorough time-resolved metabolomics study of mouse plasma over a 25-day course of *Plasmodiumchabaudi* infection. The data point to the key role of Ahr in mediating protection against acute kidney injury in a mouse model of malaria. Using untargeted metabolomics, they revealed that Ahr agonists including bilirubin, biliverdin and tryptophan metabolites are increased during infection in both mice and human patients. Using an *^/-^* mouse, they show that Ahr is necessary for protection from *Plasmodium* infection and that this protection is associated with acute kidney injury.

**Decision letter after peer review:**

Thank you for submitting your article "Metabolic profiling during malaria reveals the role of the aryl hydrocarbon receptor in regulating kidney injury" for consideration by *eLife*. Your article has been reviewed by three peer reviewers, and the evaluation has been overseen by Dominique Soldati-Favre as the Senior and Reviewing Editor. The reviewers have opted to remain anonymous.

The reviewers have discussed the reviews with one another and the Reviewing Editor has drafted this decision to help you prepare a revised submission.

All three agree that this is a major step forward. However, all of them raise important points that need to be addressed in a revised version (with a point by point rebuttal letter), before a final decision can be reached. The points re detailed in the individual reviews below.

Most importantly the reviewers request to improve the presentation of the work, to have a more detailed Discussion and to consider alternatives to the model. While some additional experiments would make it a richer manuscript, I think with good edits on the manuscript, during the pandemics new experiments could be omitted.

In this context we would like to draw your attention to changes in our revision policy that we have made in response to COVID-19 (https://elifesciences.org/articles/57162). Specifically, when editors judge that a submitted work as a whole belongs in *eLife* but that some conclusions require a modest amount of additional new data, as they do with your paper, we are asking that the manuscript be revised to either limit claims to those supported by data in hand, or to explicitly state that the relevant conclusions require additional supporting data.

Summary:

This is a thorough time-resolved metabolomics study of mouse plasma over a 25-day course of *Plasmodiumchabaudi* AJ (*Pc*) infection. The authors observe that the vast majority of metabolic alterations occur around the peak of parasitemia (acute phase), between days 5 to 10 post-infection, which coincides with the peak of hemolysis. They establish some solid parallels to human pediatric malaria as far as changes in metabolism are concerned, thereby lending credence to the use of *Pc* infection in mice as an experimental model of malaria with relevance to the human disease. They report how the Ahr is necessary for mediating protection against acute kidney injury in a mouse model of *Plasmodium*. Using untargeted metabolomics, they revealed that Ahr agonists including bilirubin, biliverdin and tryptophan metabolites are increased during infection in both mice and human patients. Using an *Ahr^/-^* mouse, they show that Ahr is necessary for protection from *Plasmodium* infection and that this protection is associated with acute kidney injury.

Essential revisions:

Reviewer #1:

Some of the conclusions drawn are not yet fully supported by the data, and important questions remain to be addressed, which would increase the overall value of this interesting study.

1) Related to Figure 3: In the third paragraph of the subsection “Malaria is characterized by stages with unique immune, metabolic, and tissue damage events”, please explain why the plasma heme level is decreased in acute infection phase. The authors show, using metabolomics, that heme levels in the plasma of *Pc* infected mice during the acute phase are lower than in uninfected controls (which should have virtually 0 heme in the plasma) (Figure 1H). Later, using a formic acid method, milimolar levels of heme were detected in the plasma of *Pc* infected mice in the acute phase (Figures 4E, 5C, 6C). This is contradictory to data in Figure 1H, and there is a wealth of data showing that heme concentration is elevated in the plasma during peak of *Plasmodium* infection, presumably due to intravascular hemolysis. Were the samples used for metabolomics treated/sampled in any way that could preclude heme mass detection? Were the mass spectrometers configured/setup in a way where heme mass could be detected? Please address.

2) The causal relationship between plasma heme and RBC counts warrants further investigation, and if this is not possible under the current circumstances, a more detailed Discussion could be provided.

3) Related to Figure 4—figure supplement 3: Although the hemolysis model used showed no apparent difference in lethality between *AhR^+/+^* and *AhR^-/-^* mice, at the PHZ dose used all the *AhR^+/+^* mice succumbed and this might have masked a possible pathogenic effect of AhR deletion in this experimental model. The authors could use a sub-lethal dose of PHZ to allow the *AhR^+/+^* or *AhR^+/-^* mice to recover from the hemolysis and anemia (usually within day 12 post injection), monitor plasma heme, iron, RBCs count and kidney injury. If the *AhR^+/-^* mice have a heme/iron recycling defect there should be an increase in heme accumulation in plasma, presumably explaining a similar observation during *Pc* infection.

4) Related to Figure 6: The authors conclude that AhR expression in the endothelium is responsible for the protective effect of AhR in *Pc* infection, based on bone marrow chimera and conditional AhR KO experiments, used to rule out hematopoietic (radiosensitive) cells, and to identify Tie2-expressing cells as responsible for the protection effect. Nevertheless, Tie2 also drives the expression of the Cre recombinase in subsets of yolk-derived tissue-resident macrophages (see https://www.ncbi.nlm.nih.gov/pmc/articles/PMC2938310/), which are likely radioresistant and therefore cannot be ruled out entirely. The authors should address this point. Better characterisation of the cells expressing AhR, for example in the kidney, would be useful to address where the expression of this receptor may be operational to prevent the onset of AKI.

Reviewer #2:

In previous work by other groups, it's been suggested that heme increases in plasma and urine during *Plasmodium* infection in mice and that this increase in heme causes AKI. In the work presented by Schneider and colleagues, they show that heme plasma levels decrease and the metabolites increase. In their Ahr model, they do see an increase in plasma (but not urine) heme at day 8 post infection, but given the kinetics of this in relation to the AKI in their model, I suggest that the heme accumulation occurs secondary to AKI and possibly as a result of the AKI, and not that heme accumulation drives AKI. The authors did a very detailed analysis on characterizing heme. They do not see increased heme in urine, they do not see increased iron containing deposits in kidneys, they see basically no differences in Hmox or Fth in the kidney and no difference in plasma heme until after the AKI appears. The authors conclude that Ahr protects from AKI by detoxifying heme, but I do not think their data support this model. What this indicates is that the authors have potentially revealed a novel relationship between heme metabolism, Ahr and AKI in the context of a *Plasmodium* infection. This is important because most of the work in the *Plasmodium* field forces the model that heme detoxification is the answer for every *Plasmodium* phenotype relating to heme. This work suggests something different and exciting.

One possibility that the authors do not seem to consider is that Ahr is necessary for a tissue protective response that protects from AKI. For example, bilirubin, biliverdin and Ahr all exhibit antioxidant properties. One possibility is that the heme metabolites are necessary to signal through Ahr to promote these antioxidant responses that protect from AKI. Similarly, this signaling pathway may be mediating anti-inflammatory responses that protect from AKI. There are likely clues to possible tissue protective responses in their RNAseq data. Is there anything in their data set to suggest such a model?

In summary, I think the data are very interesting and robust, but I would like to see the authors consider the presentation of their work and whether there is a model that better fits with the data they presented. I think there is.

Reviewer #3:

In this manuscript by Lissner et al., authors share the results of an extensive well-executed analysis on the role of AHR receptors in progression of malaria. The project has a large scale and a broad scope encompassing genetic, metabolic, clinical, histopathologic and cellular studies. While this generates massive amounts of data that will be useful for researchers in this field, often times authors lose focus moving from one set of experiments to another and the continuity of story gets disrupted. Authors' claim about the kidney protective role of AhR is very convincing with carefully carried out knock out, heterozygous and WT experiments. On the other hand, majority of metabolomic analyses only point out the consequences of the known fact that malaria generates hemolysis and heme is released as a by-product. Furthermore, comparisons of CM and *Pc* is misleading and should be avoided as I mentioned in detail in relevant section of my comments. Endothelial cell based AHR expression being critical in AHR mediated protection is the most interesting section of the paper as it finally points out the attention to a single focus so it would have been nice if authors went deeper following that lead. This paper would benefit from a rewrite connecting the loose ends into a continuous story.

Introduction, second paragraph: I recommend using WHO data to refer to the up to date global disease burden of malaria rather than a manuscript published in 2012. WHO annually releases malaria reports (world malaria report) and these may provide more up to data and more widely accepted numerical values.

Introduction, second paragraph: I recommend removing the vague statement: "much is known about how immune responses affect malaria outcome".

Introduction, fourth paragraph: Referring *Plasmodiumberghei* as rodent malaria parasite that causes lethal cerebral malaria, while not incorrect, might be misleading. The only reference that mentions *P. berghei* among cited after this sentence is Brant et al. and there, it is used as *P. berghei* ANKA which is the sub-strain that causes Experimental Cerebral Malaria (ECM) in susceptible mice. *P. berghei* NK65, another sub-strain that is very closely related *P. berghei* ANKA never results in ECM. Please use the full strain name *P. berghei* ANKA.

Figure 1A: While no revision is required, hemoglobin measurements instead of RBC counts is a better way to monitor anemia.

Subsection “Malaria is characterized by stages with unique immune, metabolic, and tissue damage events”, fourth paragraph and Figure 1I-J: Comparison of *Pc* with human CM patients is not substantiated. First of all, *Pc* is an uncomplicated malaria causing rodent parasite. CM on the other hands is the most lethal complication for both human and mouse (ECM) malaria. I recommend removing this data set since a more reasonable comparison would have been non-complicated human *Pf* vs. mouse *Pc* or Human CM vs. mouse Pb ANKA. Unless authors wish to offer these comparisons instead, there is no value of *Pc* vs. CM comparison other than that they both cause hemolysis simply because of being intra RBC microorganisms that are destroying RBCs in order to spread. *Pc* is never used to recapitulate human CM. This parasite only recapitulates the immunity development angle of human malaria with antibodies and such. This entire paragraph and relevant figures need to be reworded. Rest of Figure 1 is informative and well carried out experiments.

Subsection “AhR ligands are more abundant during acute infection”, first paragraph: Please change to Pb ANKA (refer to my comment above).

Subsection “AhR signaling is protective during *Pc* infection”, first paragraph: This is an interesting finding. Gender bias has always been mentioned in mouse malaria experiments but has not been properly documented. This is a useful information.

Figure 4: Authors should discuss why they only focused kidney but not the other organs. Spleen, lung and liver may show damages related to sequestration, intravascular hemolysis, side products, cytokines and anemia. This needs to be discussed.

Figures 5 and 6: Figure 5 does not add much to the story since it only says that radiosensitive cells do not contribute to the AhR mediated protection. Yet it still sets the stage for Figure 6. I would shorten that part and put more emphasis on Figure 6. I found Figure 6 to be very informative focusing the attention onto endothelial cells. The endothelial cell specific knock out really has potential to generate high quality original data, however authors cut that story short by only limiting the analyses to 4 major parameters. Extending that section with more sophisticated analyses would make the manuscript more complete.

Materials and methods section: age of a mouse has a significant impact on disease progression. Therefore, age matching is critical. Authors should include a statement regarding to age matching practice between control and experimental groups, otherwise 8-12 weeks is a relatively broad range for this particular experimental model.

---

## [Author Response]

Essential revisions:Reviewer #1:Some of the conclusions drawn are not yet fully supported by the data, and important questions remain to be addressed, which would increase the overall value of this interesting study.1) Related to Figure 3: In the third paragraph of the subsection “Malaria is characterized by stages with unique immune, metabolic, and tissue damage events”, please explain why the plasma heme level is decreased in acute infection phase. The authors show, using metabolomics, that heme levels in the plasma of Pc infected mice during the acute phase are lower than in uninfected controls (which should have virtually 0 heme in the plasma) (Figure 1H). Later, using a formic acid method, milimolar levels of heme were detected in the plasma of Pc infected mice in the acute phase (Figures 4E, 5C, 6C). This is contradictory to data in Figure 1H, and there is a wealth of data showing that heme concentration is elevated in the plasma during peak of Plasmodium infection, presumably due to intravascular hemolysis. Were the samples used for metabolomics treated/sampled in any way that could preclude heme mass detection? Were the mass spectrometers configured/setup in a way where heme mass could be detected? Please address.

We apologize for a typo in the manuscript – the line should read “…acute *Pc* malaria is characterized by stable plasma heme levels…” and has been fixed. Figure 1H demonstrates that heme levels don’t decrease until late infection. Apart from our error, the reviewer also notes discrepancies in heme concentrations between figures. In Figure 1H, we report plasma heme from uninfected and infected wild-type C57BL/6 mice; levels in infected mice are statistically comparable to uninfected mice during acute infection. Figures 4E, 5C, and 6C quantify plasma heme in control and *AhR^-/-^* mice. In control mice, plasma heme remains similar to baseline levels (Figures 4E, 6C). Only in susceptible *AhR^-/-^* mice, *AhR^Tie2Δ/Δ^* mice, or chimeric mice with *AhR^-/-^* radioresistant cells does plasma heme reach the millimolar levels pointed out by the reviewer. The appropriate comparison is infected mice from Figure 1H with infected control mice from the later figures (*AhR^+/-^* mice, *AhR^fl/fl^* mice, and chimeric mice with *AhR^+/-^* radioresistant cells). This comparison reveals that control or wild-type mice from Figures 1H, 4E, 5C, and 6C all have heme levels similar to uninfected mice during acute infection. We do not see a contradiction in these data.

The reviewer also mentions that heme concentration is elevated in plasma during the peak of *Plasmodium* infection. While plasma heme concentration surely varies by model, our data indicates that increased plasma heme is not a universal feature of malaria—indeed, acute *Pc* infection in C57BL/6 mice does not lead to significantly increased plasma heme. We speculate that uncomplicated malaria may not lead to increased plasma heme; rather, plasma heme increases may be a cause and/or symptom of complicated malaria. Our study demonstrates that increased plasma heme during malaria can lead to pathology and we believe that the factors underlying this increase merit further study.

2) The causal relationship between plasma heme and RBC counts warrants further investigation, and if this is not possible under the current circumstances, a more detailed Discussion could be provided.

We agree with the reviewer that the relationship between RBC density and plasma heme is ripe for exploration. RBC counts and plasma heme are certainly related, as hemolysis releases heme into plasma. The relationship is not linear, however, since plasma heme levels are affected by both heme release into plasma and the multiple processes that remove heme from plasma, such as intracellular heme metabolism by HO-1. These pathways minimize plasma heme in control *AhR^+/-^* mice during *Pc* infection (Figure 4E), despite substantial decreases in RBC density relative to baseline (Figure 3C). We believe the reviewer is interested in why this relationship breaks down in *AhR^-/-^* mice, an interest that we share.

We did not observe a large defect in heme metabolism in *AhR^-/-^* mice, through gene expression, protein analyses, and Phz-induced heme overload (Figure 4—figure supplement 2C, D, Figure 4—figure supplement 3). As the reviewer points out later, *Pc*-infected *AhR^-/-^* mice fail to upregulate *Hrg1* and *Slc40a1* in the liver, two heme transporters which may affect plasma heme levels (Figure 4—figure supplement 2D). On the other hand, if AhR-dependent regulation of these genes affected heme overload, Phz treatment should cause different plasma heme levels or sensitivity in *AhR^+/-^* and *AhR^-/-^* mice, which we did not observe (Figure 4—figure supplement 3). We did find that *AhR^-/-^* mice have slightly decreased RBC density than controls (Figure 3C); however, we suspect that this relatively minor difference is unlikely to cause the substantially increased plasma heme (Figure 4E).

We agree with the reviewer that the relationship between plasma heme and hemolysis during malaria is critical for the field to sort out. We have added clarifying sentences to the Discussion, where we propose several models that discuss hemolysis, plasma heme, and AKI (Discussion, fifth paragraph). We have also proposed an additional model, as mentioned by reviewer #2, that our data may also support (Discussion, sixth paragraph). We agree with the reviewers that further experimentation will be required differentiate between these possibilities, but believe that these studies are outside the scope of this manuscript.

3) Related to Figure 4—figure supplement 3: Although the hemolysis model used showed no apparent difference in lethality between AhR^+/+^ and AhR^-/-^ mice, at the PHZ dose used all the AhR^+/+^ mice succumbed and this might have masked a possible pathogenic effect of AhR deletion in this experimental model. The authors could use a sub-lethal dose of PHZ to allow the AhR^+/+^ or AhR^+/-^ mice to recover from the hemolysis and anemia (usually within day 12 post injection), monitor plasma heme, iron, RBCs count and kidney injury. If the AhR^+/-^ mice have a heme/iron recycling defect there should be an increase in heme accumulation in plasma, presumably explaining a similar observation during Pc infection.

The reviewer is right that the lethal dose we chose may obscure the effect of AhR loss on Phz sensitivity. However, we selected this dose because it results in plasma heme levels that are comparable to what we observe in *AhR^-/-^* mice infected with *Pc*, and at a similar time scale (plasma heme increases to above 1000 μm within one day). These features, we reasoned, faithfully recapitulate the hemolysis caused by malaria. While a lower dose would cause less lethality, it would also not model this aspect of malaria. Furthermore, we monitored several factors in response to Phz treatment in addition to survival, including plasma heme and kidney function, and did not see a difference in susceptibility in any metric. Due to these reasons, and experimental challenges imposed by the pandemic, we did not perform the experiment the reviewer suggested.

4) Related to Figure 6: The authors conclude that AhR expression in the endothelium is responsible for the protective effect of AhR in Pc infection, based on bone marrow chimera and conditional AhR KO experiments, used to rule out hematopoietic (radiosensitive) cells, and to identify Tie2-expressing cells as responsible for the protection effect. Nevertheless, Tie2 also drives the expression of the Cre recombinase in subsets of yolk-derived tissue-resident macrophages (see https://www.ncbi.nlm.nih.gov/pmc/articles/PMC2938310/), which are likely radioresistant and therefore cannot be ruled out entirely. The authors should address this point. Better characterisation of the cells expressing AhR, for example in the kidney, would be useful to address where the expression of this receptor may be operational to prevent the onset of AKI.

We agree with the reviewer on this point and originally alluded to the fact that tissue-resident macrophages also fit the criteria of radioresistant *Tie2*-expressing cells. While future experiments to identify the *Tie2*-expressing radioresistant cells at play in this phenotype, we believe that such work is outside of the scope of this manuscript and should be addressed in future studies. However, we agree with the reviewer that clarification around this issue is necessary in the manuscript. We have provided more detail in the subsection “AHR is necessary in *Tek*-expressing cells to control parasitemia, plasma heme, and AKI during *Pc* infection” and throughout the Discussion.

Reviewer #2:[…] One possibility that the authors do not seem to consider is that Ahr is necessary for a tissue protective response that protects from AKI. For example, bilirubin, biliverdin and Ahr all exhibit antioxidant properties. One possibility is that the heme metabolites are necessary to signal through Ahr to promote these antioxidant responses that protect from AKI. Similarly, this signaling pathway may be mediating anti-inflammatory responses that protect from AKI. There are likely clues to possible tissue protective responses in their RNAseq data. Is there anything in their data set to suggest such a model?In summary, I think the data are very interesting and robust, but I would like to see the authors consider the presentation of their work and whether there is a model that better fits with the data they presented. I think there is.

We think the reviewer has raised a very insightful point. In the first model we proposed, AhR functions exclusively in pathogen control, and we have added sentences to clarify this point (Discussion, fifth paragraph). This model suggests that, in the absence of AhR, parasitemia is increased as early as 4 DPI (Figure 3A), leading to more hemolysis in *AhR^-/-^* mice (Figure 3C). Despite equivalent plasma heme concentrations, intracellular heme concentrations could nevertheless be elevated in renal proximal tubule epithelial cells, which import and metabolize heme during malaria and are sensitive to heme toxicity (Ramos, 2019). This could explain impaired kidney function at 7 DPI (Figure 4B) despite equivalent plasma heme (Figure 4E).

In the second model we considered, AhR functions in both pathogen control and heme metabolism. We believe that our data do not support this model, as we did not detect any major deficits in heme metabolism in *AhR^-/-^* mice. Nevertheless, we remained unable to pinpoint the source of such extremely elevated plasma heme (Discussion, sixth paragraph).

The reviewer suggests a third possibility in which AhR is required for both pathogen control and tissue protection in the kidney. This model orients elevated plasma heme as a consequence, rather than the initial cause, of impaired kidney function. Particularly tantalizing is the role of the heme metabolites bilirubin and biliverdin in activating AhR, suggesting a possible feedback loop in which AhR is activated by heme metabolism to protect from heme toxicity. As an important caveat to this model, impaired kidney function is observed prior to elevated plasma heme in *AhR^-/-^* mice (Figure 4B, E), but not in *AhR^Tie2Δ/Δ^* mice (Figure 6C, D) or chimeric mice with *AhR^/-^* radioresistant cells (Figure 5, longitudinal data not shown). Whether this is biologically relevant or simply an artifact of different mouse models will have important implications, since the relative timing of heme elevation and kidney impairment is critical. While we do not believe that our RNA-seq data, a time-course analysis of wild-type livers during *Pc* infection, provide insight into this pathway, we agree that the role of kidney-intrinsic AhR signaling during *Pc* infection merits further investigation. We are grateful to the reviewer for this insight and have included further discussion of each model in the Discussion. Additionally, when we previously referred to “heme-mediated AKI” in the manuscript, we now refer to the development of high plasma heme and AKI without any causal implications.

Reviewer #3:In this manuscript by Lissner et al., authors share the results of an extensive well-executed analysis on the role of AHR receptors in progression of malaria. The project has a large scale and a broad scope encompassing genetic, metabolic, clinical, histopathologic and cellular studies. While this generates massive amounts of data that will be useful for researchers in this field, often times authors lose focus moving from one set of experiments to another and the continuity of story gets disrupted. Authors' claim about the kidney protective role of AhR is very convincing with carefully carried out knock out, heterozygous and WT experiments. On the other hand, majority of metabolomic analyses only point out the consequences of the known fact that malaria generates hemolysis and heme is released as a by-product. Furthermore, comparisons of CM and Pc is misleading and should be avoided as I mentioned in detail in relevant section of my comments. Endothelial cell based AHR expression being critical in AHR mediated protection is the most interesting section of the paper as it finally points out the attention to a single focus so it would have been nice if authors went deeper following that lead. This paper would benefit from a rewrite connecting the loose ends into a continuous story.Introduction, second paragraph: I recommend using WHO data to refer to the up to date global disease burden of malaria rather than a manuscript published in 2012. WHO annually releases malaria reports (world malaria report) and these may provide more up to data and more widely accepted numerical values.

We thank the reviewer for this insight and have made the change suggested.

Introduction, second paragraph: I recommend removing the vague statement: "much is known about how immune responses affect malaria outcome".

We have made the suggested change.

Introduction, fourth paragraph: Referring Plasmodium berghei as rodent malaria parasite that causes lethal cerebral malaria, while not incorrect, might be misleading. The only reference that mentions P. berghei among cited after this sentence is Brant et al. and there, it is used as P. berghei ANKA which is the sub-strain that causes Experimental Cerebral Malaria (ECM) in susceptible mice. P. berghei NK65, another sub-strain that is very closely related P. berghei ANKA never results in ECM. Please use the full strain name P. berghei ANKA.

The reviewer raises a very good point and we have made this change throughout the manuscript.

Figure 1A: While no revision is required, hemoglobin measurements instead of RBC counts is a better way to monitor anemia.

We understand the reviewer’s point and agree that hemoglobin measurements offer complementary information.

Subsection “Malaria is characterized by stages with unique immune, metabolic, and tissue damage events”, fourth paragraph and Figure 1I-J: Comparison of Pc with human CM patients is not substantiated. First of all, Pc is an uncomplicated malaria causing rodent parasite. CM on the other hands is the most lethal complication for both human and mouse (ECM) malaria. I recommend removing this data set since a more reasonable comparison would have been non-complicated human Pf vs. mouse Pc or Human CM vs. mouse Pb ANKA. Unless authors wish to offer these comparisons instead, there is no value of Pc vs CM comparison other than that they both cause hemolysis simply because of being intra RBC microorganisms that are destroying RBCs in order to spread. Pc is never used to recapitulate human CM. This parasite only recapitulates the immunity development angle of human malaria with antibodies and such. This entire paragraph and relevant figures need to be reworded. Rest of Figure 1 is informative and well carried out experiments.

We thank the reviewer for this point. We agree that uncomplicated malaria caused by *Pc* and CM caused by *P. falciparum* are considerably different types of malaria and cannot be substituted for each other. In this sense, it is surprising that the plasma metabolomes are so similar (Figure 1I)! On the other hand, *Pc* and *Pf* are closely-related species that cause infections with many similar features, such as massive RBC destruction and systemic immune activation, as the reviewer points out.

Our mouse model of *Pc-*infected wild-type mice is most similar to uncomplicated *Pf* malaria, and the reviewer correctly suggests that an uncomplicated malaria dataset would be best for our comparative analysis. However, analysis is challenged by the fact that large untargeted metabolomics datasets are relatively rare, compared to targeted analysis; and when untargeted metabolomics analysis has been employed in the past, technical limitations at the time resulted in the identification of relatively few metabolites. For example, the most detailed dataset meeting the reviewer’s specifications that we have examined focused on uncomplicated *Pf-* or *P. Vivax* infected patients (found under MTBLS664). That study identified less than a third of the molecules found in our study (190 vs. 587 metabolites), and none of the heme-related metabolites that we discuss.

Metabolomics datasets that capture the ideal type of malaria in adequate detail remain elusive. For this reason, we performed our comparative analysis using metabolomics data from CM patients, which identified 432 metabolites. The reviewer is correct that a more ideal comparison would be preferable. However, given the limited data availability and the general similarities of malaria caused by all *Plasmodium* species, we disagree that the comparison is inappropriate. We have added a sentence to clarify the limitations of our analysis in the subsection “Malaria is characterized by stages with unique immune, metabolic, and tissue damage event”.

Subsection “AhR ligands are more abundant during acute infection”, first paragraph: Please change to Pb ANKA (refer to my comment above).

We have made this change.

Subsection “AhR signaling is protective during Pc infection”, first paragraph: This is an interesting finding. Gender bias has always been mentioned in mouse malaria experiments but has not been properly documented. This is a useful information.

We agree with the reviewer that gender bias in malaria is a rich area for future research.

Figure 4: Authors should discuss why they only focused kidney but not the other organs. Spleen, lung and liver may show damages related to sequestration, intravascular hemolysis, side products, cytokines and anemia. This needs to be discussed.

We agree with the reviewer’s point that other organs may also be relevant to this phenotype. For that reason, we originally analyzed all tissues mentioned by the reviewer—spleen, lung, and liver. We do mention that our histological analysis of liver tissue revealed less severe damage in *AhR^-/-^* mice (subsection “AHR signaling is protective during *Pc* infection”). We have now added a sentence to indicate that spleen and lung tissue showed no significant differences between genotypes.

Figures 5 and 6: Figure 5 does not add much to the story since it only says that radiosensitive cells do not contribute to the AhR mediated protection. Yet it still sets the stage for Figure 6. I would shorten that part and put more emphasis on Figure 6. I found Figure 6 to be very informative focusing the attention onto endothelial cells. The endothelial cell specific knock out really has potential to generate high quality original data, however authors cut that story short by only limiting the analyses to 4 major parameters. Extending that section with more sophisticated analyses would make the manuscript more complete.

We agree that the role of AhR in *Tie2*-expressing cells merits further study. Thorough characterization of this phenotype, however, requires extensive experimentation and will likely warrant an entire story in the future. We feel that additional experimentation on this point is outside the scope of this paper.

Materials and methods section: age of a mouse has a significant impact on disease progression. Therefore, age matching is critical. Authors should include a statement regarding to age matching practice between control and experimental groups, otherwise 8-12 weeks is a relatively broad range for this particular experimental model.

We appreciate the reviewer’s point. For this reason, we have always carefully matched ages. By using littermates that fall within the 8-12 week old range, we ensure that our experiments have age-matched control and experimental mice. That said, we have not seen a difference in disease severity between mice that are 8 weeks versus 12 weeks old. To clarify this point in the manuscript, we have moved all discussion of mouse age to the subsection “Cross-sectional infection”.